# PARQ: Piecewise-Affine Regularized Quantization

Lisa Jin [1]   Jianhao Ma [2]   Zechun Liu [3]   Andrey Gromov [1]   Aaron Defazio [1]   Lin Xiao [1]

## Abstract

We develop a novel optimization method for quantization-aware training (QAT). Specifically, we show that *convex*, piecewise-affine regularization (PAR) can effectively induce the model parameters to cluster towards discrete, quantized values. We minimize PAR-regularized loss functions using an *aggregate proximal* stochastic gradient method (AProx) and show that it enjoys *last-iterate convergence*. Our approach provides an interpretation of the straight-through estimator (STE), a widely used heuristic for QAT, as the asymptotic form of PARQ. We present numerical experiments to demonstrate that PARQ obtains competitive performance on convolution- and transformer-based vision tasks.

## 1. Introduction

Modern deep learning models exhibit exceptional vision and language processing capabilities, but often come with excessive sizes and demands on memory and computing. Quantization is an effective approach for model compression, which can significantly reduce their memory footprint, computing cost, as well as the latency for inference (e.g., Han et al., 2016; Sze et al., 2017). There are two main classes of quantization methods: post-training quantization (PTQ) and quantization-aware training (QAT). Both are widely adopted and receive extensive research; see the recent survey papers by Gholami et al. (2022) and Fournarakis et al. (2022) and references therein.

PTQ converts the weights of a pre-trained model directly to lower precision without repeating the training pipeline; thus it has less overhead and is relatively easy to apply (Nagel et al., 2020; Cai et al., 2020; Chee et al., 2024). However, it is limited mainly to 4 or more bit regimes and can suffer steep performance drops with fewer bits (Yao et al., 2022; Dettmers & Zettlemoyer, 2023). This is especially the case for transformer-based models, which are more difficult to quantize (Bai et al., 2021; Qin et al., 2022) compared to convolutional architectures (Martinez et al., 2019; Qin et al., 2020). On the other hand, QAT integrates quantization into pre-training and/or fine-tuning processes and can produce low-bit (including binary) models with mild performance degradation (e.g. Fan et al., 2021; Liu et al., 2022).

A key ingredient of QAT is the so-called *straight-through estimator* (STE), which was proposed as a heuristic (Bengio et al., 2013; Courbariaux et al., 2015) and has been extremely successful in practice (e.g., Rastegari et al., 2016; Hubara et al., 2018; Esser et al., 2019). Many efforts have been made to demystify the effectiveness of STE, especially through the lens of optimization algorithms (e.g., Li et al., 2017; Yin et al., 2018; 2019; Bai et al., 2019; Ajanthan et al., 2021; Dockhorn et al., 2021; Lu et al., 2023). However, significant gaps remain between theory and practice.

In this paper, we develop a principled method for QAT based on *convex* regularization and interpret STE as the asymptotic form of an *aggregate proximal* stochastic gradient method. The convex regularization framework admits stronger convergence guarantees than previous work and allows us to prove the *last-iterate convergence* of the method.

### 1.1. The Straight-Through Estimator (STE)

We consider training a machine learning model with parameters $w \in \mathbf{R}^d$ and let $f(w, z)$ denote the loss of the model on a training example $z$. Our goal is to minimize the population loss $f(w) = \mathbf{E}_z[f(w, z)]$ where $z$ follows some unknown probability distribution. Here, we focus on the classical stochastic gradient descent (SGD) method. During each iteration of SGD, we draw a random training example (or mini-batch) $z^t$ and update the model parameter as

$$w^{t+1} = w^t - \eta_t \nabla f(w^t, z^t), \qquad (1)$$

where $\nabla f(\cdot, z^t)$ denotes the stochastic gradient with respect to the first argument (here being $w^t$) and $\eta_t$ is the step size.

QAT methods modify SGD by adding a quantization step. In particular, the BinaryConnect method (Courbariaux et al., 2015) can be written as

$$u^{t+1} = u^t - \eta_t \nabla f(Q(u^t), z^t), \qquad (2)$$

where $Q(\cdot)$ is the coordinate-wise projection onto the set

[1]Meta FAIR, United States. [2]Dept. of Industrial and Operational Engineering, University of Michigan, Ann Arbor, MI, United States. [3]Meta Reality Labs, United States. Correspondence to: Lisa Jin <lvj@meta.com>, Lin Xiao <linx@meta.com>.

*Proceedings of the 42nd International Conference on Machine Learning*, Vancouver, Canada. PMLR 267, 2025. Copyright 2025 by the author(s).

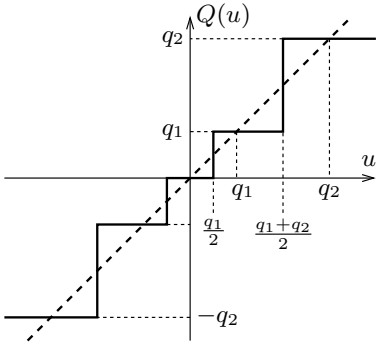

*Figure 1.* A quantization map with $\mathcal{Q} = \{0, \pm q_1, \pm q_2\}$.

$\{\pm 1\}^d$. It readily generalizes to projection onto $\mathcal{Q}^d$ where $\mathcal{Q}$ is a finite set of arbitrary quantization values. Figure 1 shows an example with $\mathcal{Q} = \{0, \pm q_1, \pm q_2\}$.

Notice that in Equation (2) we switched the notation from $w^t$ to $u^t$, because we would like to define $w^t = Q(u^t)$ as the quantized model parameters. This reveals a key feature of QAT: the stochastic gradient in (2) is computed at $w^t$ instead of $u^t$ itself (which would be equivalent to (1)). Here we regard $u^t$ as a full-precision latent variable that is used to accumulate the gradient computed at $w^t$, and the quantization map $Q(\cdot)$ is applied to the latent variable $u^{t+1}$ to generate the next quantized variable $w^{t+1}$.

The notion of STE arises from the intent of computing an approximate gradient of the loss function with respect to $u^t$. Let us define the function $\tilde{f}(u, z) := f(Q(u), z) = f(w, z)$ in light of $w = Q(u)$. Then we have for each $i = 1, \ldots, d$,

$$\frac{\partial \tilde{f}}{\partial u_i} = \frac{\partial f}{\partial w_i} \frac{dw_i}{du_i} = \frac{\partial f}{\partial w_i} \frac{dQ(u_i)}{du_i}.$$

However, due to the staircase shape of the quantization map, we have $dQ(u_i)/du_i = 0$ and thus $\nabla \tilde{f}(u, z) = 0$ almost everywhere, which prevent effective learning. In order to fix this problem, STE tries to "construct" a nontrivial gradient with respect to $u$, by simply treating $Q(\cdot)$ as the identity map during backpropagation, i.e., replacing $dQ(u_i)/du_i$ with 1 in the above equation. This leads to the "straight-through" approximation

$$\nabla \tilde{f}(u, z) \overset{\text{STE}}{\approx} \nabla f(w, z) = \nabla f(Q(u), z),$$

so that one can interpret Equation (2) as an (approximate) SGD update for minimizing the composite function $\tilde{f}(u)$.

There are several issues with this argument. First, we know exactly that $dQ(u_i)/du_i = 0$ almost everywhere, so there is no need for "approximation." Second, any approximation that replaces 0 with 1 in this context warrants scrutiny of the resulting bias and the consequences on training stability. Existing works on this are restricted to special cases and weak convergence results (Li et al., 2017; Yin et al., 2019).

Alternatively, we can view (2) as an implicit algorithm for updating $w^t$ and analyze its convergence. More explicitly,

$$\begin{aligned} u^{t+1} &= u^t - \eta_t \nabla f(w^t, z^t), \\ w^{t+1} &= Q(u^{t+1}). \end{aligned} \tag{3}$$

Here $u^t$ serves as an auxiliary variable that accumulates past gradients evaluated at $w^0, \ldots, w^t$ (similar to momentum). This formulation allows application of the powerful framework of regularization and proximal gradient methods (e.g., Bai et al., 2019; Dockhorn et al., 2021). And this is the path we take in this paper.

### 1.2. Outline and contributions

In Section 2, we review the framework of regularization and introduce a family of *convex*, piecewise-affine regularizers (PAR). In addition, we derive the first-order optimality conditions for minimizing PAR-regularized functions.

In Section 3, we derive an aggregate proximal gradient method (AProx) for solving PAR-regularized minimization problems and provide its convergence analysis for convex losses. AProx applies a soft-quantization map that evolves over the iterations and asymptotically converges to hard quantization, thus giving a principled interpretation of STE.

In Section 4, we present PARQ (Piecewise-Affine Regularized Quantization), a practical implementation of AProx with PAR regularization that does not need to pre-determine the quantization values and regularization strength.

In Section 5, we conduct QAT experiments on low-bit quantization of convolution- and transformer-based vision models and demonstrate that PARQ obtains competitive performance compared to STE/BinaryConnect, as well as other methods based on nonconvex regularization.

We note that Dockhorn et al. (2021) already used the regularization framework and proximal optimization to interpret (demystify) BinaryConnect and developed a generalization called ProxConnect. In fact, AProx is equivalent to Prox-Connect albeit following quite different derivations. Nevertheless, we make the following novel contributions.

- We propose *convex* PAR to induce quantization. Dockhorn et al. (2021) focused on monotone (non-decreasing) proximal maps, which can correspond to arbitrary regularization. Although they presented convergence results for convex regularization, no such example was given to demonstrate its relevance. Beyond closing this gap between theory and practice, our construction of convex PAR is rather surprising counterintuitive for the purpose of quantization.

- We derive first-order optimality conditions for minimizing PAR-regularized functions. They reveal the critical role of *nonsmoothness* in inducing quantization.

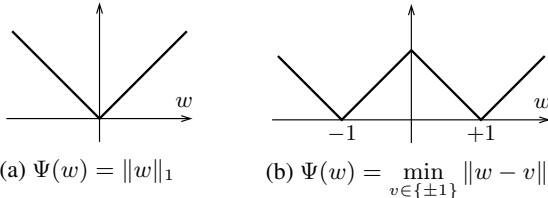

*Figure 2.* Illustration of two nonsmooth regularizers.

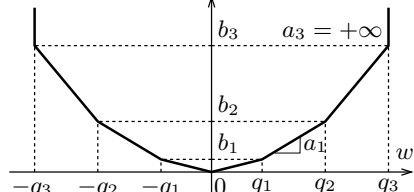

*Figure 3.* Convex PAR: $\Psi(w) = \max_k\{a_k(|w| - q_k) + b_k\}$.

- We prove *last-iterate convergence* of AProx. The convergence results of Dockhorn et al. (2021) concern the averaged iterates generated by ProxConnect/AProx. While such results are conventional in the stochastic optimization literature, they are far from satisfactory for QAT, because the averaged iterate may not be quantized even if every iterate is quantized. Last-iterate convergence gives a much stronger guarantee.

- We propose a practical implementation called PARQ that can adaptively choose the quantization values and regularization strength in an online fashion.

Our implementation of PARQ in PyTorch is available at https://github.com/facebookresearch/parq.

## 2. Piecewise affine regularization (PAR)

Regularization is a common approach for inducing desired properties of machine learning models, by minimizing a weighted sum of the loss function $f$ and a regularizer $\Psi$:

$$\underset{w \in \mathbf{R}^d}{\text{minimize}} \quad f(w) + \lambda\Psi(w), \qquad (4)$$

where $\lambda \in \mathbf{R}_+$ is a parameter to balance the relative strength of regularization. For example, it is well known that $L_2$-regularization helps generalization by preferring smaller model parameters, and $L_1$-regularization, illustrated in Figure 2(a), induces sparsity (e.g., Hastie et al., 2009).

There have been many attempts of using regularization to induce quantization (e.g., Carreira-Perpiñán & Idelbayev, 2017; Yin et al., 2018; Bai et al., 2019). An obvious choice is to let $\Psi$ be the indicator function of $\mathcal{Q}^d$; in other words, $\Psi(w) = \sum_{i=1}^d \delta_{\mathcal{Q}}(w_i)$ where

$$\delta_{\mathcal{Q}}(w_i) = \begin{cases} 0 & \text{if } w_i \in \mathcal{Q}, \\ +\infty & \text{otherwise.} \end{cases} \qquad (5)$$

Then minimizing $f(w) + \lambda\Psi(w)$ is equivalent to the constrained optimization problem of minimizing $f(w)$ subject to $w \in \mathcal{Q}^d$, which is combinatorial in nature and very hard to solve in general. Yin et al. (2018) propose to use the Moreau envelope of the indicator function, which under the Euclidean metric gives $\Psi(w) = \min_{v \in \mathcal{Q}^d} \|v - w\|_2^2$. A nonsmooth version is proposed by Bai et al. (2019) under

the $L_1$-metric, resulting in $\Psi(w) = \min_{v \in \mathcal{Q}^d} \|v - w\|_1$; Figure 2(b) shows a W-shaped example in one dimension.

We argue that the effectiveness of a regularizer for inducing quantization largely relies on two critical properties: *nonsmoothness* and *convexity*. Smooth regularizers such as $\text{dist}(w, \mathcal{Q}^d) := \min_{v \in \mathcal{Q}^d} \|v - w\|_2^2$ behave like $\|w\|_2^2$ locally, thus do not induce zero or any discrete structure. On the other hand, nonsmooth regularizers behave like $\|w\|_1$ near zero, so they can trap model parameters at the set of nondifferentiable points—more suitable for quantization.

Convexity concerns the global behavior of regularization. For example, the popularity of $L_1$-regularization for sparse optimization is largely attributed to its convexity besides being nonsmooth. On the other hand, it is hard for a gradient-based algorithm to cross the middle hill in the nonconvex W-shaped regularizer shown in Figure 2(b), if the initial weights are trapped in the wrong valley from the optimal ones. Therefore, ideally we would like to construct a regularizer that is both nonsmooth and convex.

### 2.1. Definition of PAR

To simplify presentation, we assume $\Psi(w) = \sum_{i=1}^d \Psi(w_i)$ and use the same notation $\Psi$ for the function of a vector or one of its coordinates (it should be self-evident from the context). For most of the discussion, we focus on the scalar case and omit the subscript $i$ or simply assume $d = 1$.

Suppose that the set of target quantization values is given as $\mathcal{Q} = \{0, \pm q_1, \ldots, \pm q_m\}$ with $0 = q_0 < q_1 < \cdots < q_m$. We define a piecewise-affine regularizer (PAR) as

$$\Psi(w) = \max_{k \in \{0, \ldots, m\}}\{a_k(|w| - q_k) + b_k\}, \qquad (6)$$

where the slopes $\{a_k\}_{k=0}^m$ are free parameters that satisfy $0 \le a_0 < a_1 < \cdots < a_m = +\infty$, and $\{b_k\}_{k=0}^m$ are determined by setting $b_0 = 0$, $q_0 = 0$, and

$$b_k = b_{k-1} + a_{k-1}(q_k - q_{k-1}), \qquad k = 1, \ldots, m.$$

As shown in Figure 3, $(\pm q_k, b_k)$ are the reflection points of the piecewise-affine graph. The function $\Psi(w)$ is convex because the maximum of finite linear functions is convex (Boyd & Vandenberghe, 2004, Section 3.2.3).

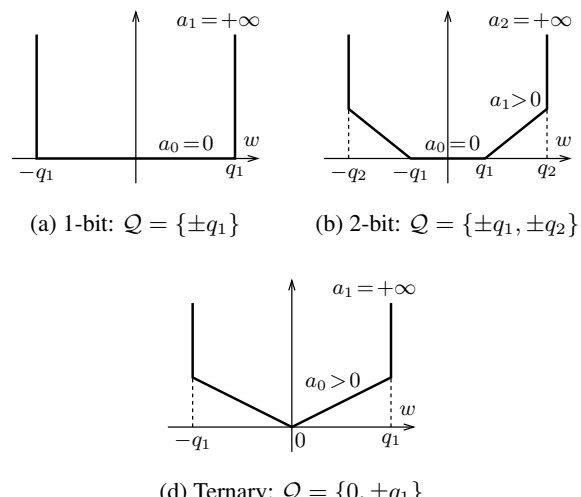

(a) 1-bit: $\mathcal{Q} = \{\pm q_1\}$    (b) 2-bit: $\mathcal{Q} = \{\pm q_1, \pm q_2\}$

(d) Ternary: $\mathcal{Q} = \{0, \pm q_1\}$

*Figure 4.* Three special cases of PAR for low-bit quantization.

We note that setting $a_0 = 0$ effectively removes $q_0 = 0$ from the quantization set $\mathcal{Q}$ because it is no longer a reflection point of $\Psi$. Figure 4 illustrates three special cases of PAR for low-bit quantization, where both Figures 4(a) and 4(b) have $a_0 = 0$. Finally, we note that the above definition of PAR is symmetric around zero, for the convenience of presentation. It is straightforward to extend to the asymmetric case.

## 2.2. Optimality conditions

In order to understand how PAR can induce quantization, we examine the optimality conditions of minimizing PAR-regularized functions. Suppose $f$ is differentiable and $w^\star$ is a solution to the optimization problem (4). The first-order optimality condition for this problem is (see, e.g., Wright & Recht, 2022, Theorem 8.18)

$$0 \in \nabla f(w^\star) + \lambda \partial \Psi(w^\star),$$

where $\partial \Psi(w^\star)$ denotes the subdifferential of $\Psi$ at $w^\star$, and $\lambda \partial \Psi(w^\star)$ means multiplying each element of the set $\partial \Psi(w^\star)$ by $\lambda$. For convenience, we rewrite it as $\nabla f(w^\star) \in -\lambda \partial \Psi(w^\star)$, which breaks down into the following cases:

$$
\begin{aligned}
w_i^\star = -q_k, &\quad \Longleftarrow \quad \nabla_i f(w^\star) \in \lambda(a_{k-1}, a_k) \\
w_i^\star \in (-q_k, -q_{k-1}) &\quad \Longrightarrow \quad \nabla_i f(w^\star) = \lambda a_{k-1} \\
w_i^\star = 0 &\quad \Longleftarrow \quad -\nabla_i f(w^\star) \in \lambda(-a_0, a_0) \\
w_i^\star \in (q_{k-1}, q_k) &\quad \Longrightarrow \quad \nabla_i f(w^\star) = -\lambda a_{k-1} \\
w_i^\star = q_k, &\quad \Longleftarrow \quad \nabla_i f(w^\star) \in \lambda(-a_k, -a_{k-1}).
\end{aligned}
$$

Here $\nabla_i$ denotes the $i$th coordinate of the vector $\nabla f$, the subscript $i$ runs from 1 through $d$, and the piecewise-affine index $k$ runs from 1 through $m$. The symbol $\Longleftarrow$ ($\Longrightarrow$) means that the expression on the left side is a necessary (sufficient) condition for the expression on the right side.

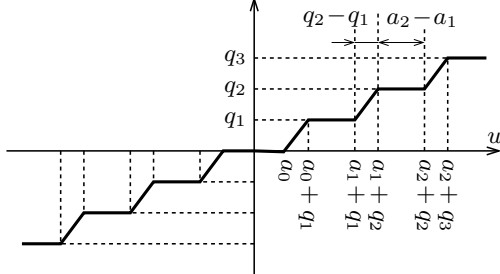

*Figure 5.* Graph of $\mathbf{prox}_\Psi(u)$.

We immediately recognize that the sufficient condition for $w_i^\star = 0$ (third equation above) is the same as for the $L_1$-regularization $\Psi(w) = \lambda \cdot a_0 \|w\|_1$. Further examination reveals that for any weight not clustered at a discrete value in $\mathcal{Q}$, i.e., if $w_i^\star \in (q_{k-1}, q_k)$ for some $k$, the corresponding partial derivative $\nabla_i f(w^\star)$ must equal the singleton $-\lambda a_{k-1}$. Conversely, almost all values of the partial derivatives of $f$, except for the $2m$ discrete values, $\{\pm \lambda a_{k-1}\}_{k=1}^m$, can be balanced by assigning the model parameters at the $2m + 1$ discrete values in $\mathcal{Q} = \{0, \pm q_1, \ldots, \pm q_m\}$. Intuitively, this implies that the model parameters at optimality are more likely to cluster at these discrete values. We will derive an algorithm that manifests this property rigorously in Section 3.

## 2.3. Proximal mapping of PAR

A fundamental tool for solving problem (4) is the *proximal map* of the regularizer $\Psi$, defined as

$$\mathbf{prox}_\Psi(u) = \arg\min_w \left\{ \Psi(w) + \tfrac{1}{2} \|w - u\|_2^2 \right\}.$$

See, e.g., Wright & Recht (2022, §8.6) for further details. For the PAR function defined in (6), its proximal map has the following closed-form solution (letting $a_{-1} = 0$)

$$
\mathbf{prox}_\Psi(u) = \begin{cases} \text{sgn}(u) q_k & \text{if } |u| \in [a_{k-1} + q_k, \ a_k + q_k], \\ u - \text{sgn}(u) a_k & \text{if } |u| \in [a_k + q_k, \ a_k + q_{k+1}]. \end{cases}
\tag{7}
$$

where $\text{sgn}(\cdot)$ denotes the sign or signum function.

Figure 5 shows the graph of $\mathbf{prox}_\Psi(u)$, which is clearly monotone non-decreasing in $u$. According to Yu et al. (2015, Proposition 3), a (possibly multi-valued) map is a proximal map of some function if and only if it is compact-valued, monotone and has a closed graph. For example, the hard-quantization map in Figure 1 is the proximal map of the (nonconvex) indicator function $\delta_\mathcal{Q}$ in (5). Dockhorn et al. (2021) work with monotone proximal maps directly without specifying the regularizer itself. In contrast, we construct a convex regularizer, and show that it can effectively induce quantization and obtain competitive performance in practice, together with stronger convergence guarantees.

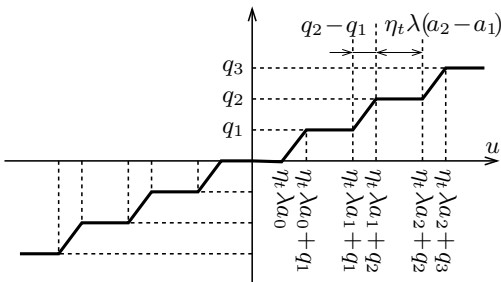

*Figure 6.* Graph of $\mathbf{prox}_{\eta_t \lambda \Psi}(u)$.

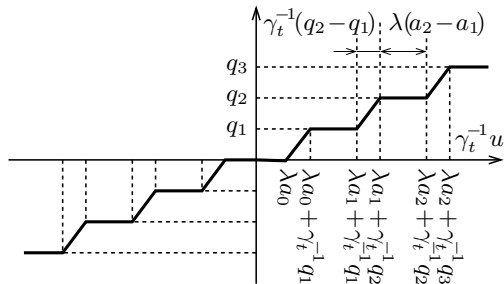

*Figure 7.* Graph of $\mathbf{prox}_{\gamma_t \lambda \Psi}(u)$ with scaled input.

## 3. The AProx Algorithm

The regularization structure of problem (4) can be well exploited by the *proximal gradient* method

$$w^{t+1} = \mathbf{prox}_{\eta_t \lambda \Psi}\left(w^t - \eta_t \nabla f(w^t)\right), \qquad (8)$$

where $\mathbf{prox}_{\eta_t \lambda \Psi}$ is the proximal map of the scaled function $\eta_t \lambda \Psi$. Since $\eta_t \lambda$ effectively scales the slopes $\{a_k\}_{k=1}^m$ (with $\mathcal{Q}$ fixed), we obtain $\mathbf{prox}_{\eta_t \lambda \Psi}$ by simply replacing $a_k$ in (7) with $\eta_t \lambda a_k$ and the corresponding map is shown in Figure 6.

If $f$ is convex and $\nabla f$ is $L$-Lipschitz continuous, then using the constant step size $\eta_t = 1/L$ leads to a convergence rate of $O(1/t)$ (e.g., Wright & Recht, 2022, Theorem 9.6).

In the context of machine learning, we minimize the expected loss over a large amount of data, i.e., $f(w) = \mathbf{E}_z[f(w, z)]$. The Prox-SGD method replaces $\nabla f(w^t)$ in (8) with the stochastic gradient $g^t := \nabla_w f(w^t, z^t)$:

$$w^{t+1} = \mathbf{prox}_{\eta_t \lambda \Psi}\left(w^t - \eta_t g^t\right). \qquad (9)$$

However, it is well known that for the (proximal) SGD method to converge, we need diminishing and non-summable step sizes (e.g., Robbins & Monro, 1951), i.e.,

$$\eta_t \to 0 \qquad \text{and} \qquad \sum_{t=1}^{\infty} \eta_t = +\infty. \qquad (10)$$

In this case, the flat segments on the graph of $\mathbf{prox}_{\eta_t \lambda \Psi}$, as shown in Figure 6, with lengths $\eta_t \lambda(a_k - a_{k-1})$, will all shrink to zero when $\eta_t \to 0$ (except at the two ends because $a_m = +\infty$). Therefore, the graph converges to the identity map clipped flat outside of $[-q_m, +q_m]$ and we lose the action of quantization. This issue parallels that of using Prox-SGD with $L_1$-regularization, which does not produce sparse solutions because of the shrinking deadzone in the soft-thresholding operator as $\eta_t \to 0$ (Xiao, 2010).

To overcome the problem of diminishing regularization, we propose AProx, an aggregate proximal stochastic gradient method. Aprox shares a similar form with BinaryConnect (Courbariaux et al., 2015). Specifically, it replaces the hard-quantization $Q(\cdot)$ in (3) with an *aggregate* proximal map:

$$\begin{aligned} u^{t+1} &= u^t - \eta_t g^t, \\ w^{t+1} &= \mathbf{prox}_{\gamma_t \lambda \Psi}(u^{t+1}), \end{aligned} \qquad (11)$$

where $\gamma_t = \sum_{s=1}^t \eta_s$. Here $\mathbf{prox}_{\gamma_t \lambda \Psi}$ is called an aggregate map because $\lambda \Psi$ is scaled by the aggregate step size $\gamma_t$. In fact, BinaryConnect is a special case of AProx with $\Psi$ being the indicator function of $\mathcal{Q}^d$ given in (5). The indicator function and its proximal map (Figure 1) is invariant under arbitrary scaling, thus hiding the subtlety of aggregation.

The graph of $\mathbf{prox}_{\gamma_t \lambda \Psi}$ can be obtained by replacing $\eta_t$ in Figure 6 with $\gamma_t$. However, according to (10), we have

$$\gamma_t = \sum_{s=1}^t \eta_s \to +\infty,$$

which implies that the flat segments in the graph, now with lengths $\gamma_t \lambda(a_k - a_{k-1})$, grow larger and larger, which is *opposite* to the Prox-SGD method. (In both cases, the sloped segments has fixed length $q_k - q_{k-1}$.)

For the ease of visualization, we rescale the input $u$ by $\gamma_t^{-1}$ and obtain the graph in Figure 7. In this scaled graph, the lengths of the flat segments $\lambda(a_k - a_{k-1})$ stay constant but the sloped segments, with lengths $\gamma_t^{-1}(q_k - q_{k-1})$, shrink as $\gamma_t$ increases. Asymptotically, as $\gamma_t \to \infty$, the graph converges to hard quantization, as shown in Figure 8.

### 3.1. AProx versus Prox-SGD and ProxConnect

To better understand the difference between AProx and Prox-SGD, we rewrite Prox-SGD in (9) as

$$\begin{aligned} u^{t+1} &= w^t - \eta_t g^t, \\ w^{t+1} &= \mathbf{prox}_{\eta_t \lambda \Psi}(u^{t+1}), \end{aligned} \qquad (12)$$

which differ from AProx in (11) in two places (highlighted in blue). Here we give an intuitive interpretation of these differences. First, notice that the objective in (4) is the sum of $f$ and $\lambda \Psi$, and both methods make progress by using the stochastic gradient of $f$ (forward step) and the proximal map of $\lambda \Psi$ (backward step) — in a balanced manner.

- In Prox-SGD, $u^{t+1}$ is a combination of $w^t$ and $-\eta_t g^t$. But $w^t$ already contains contributions from both $f$ and $\lambda \Psi$, through $\{-\eta_s g^s\}_{s=1}^{t-1}$ and $\{\mathbf{prox}_{\eta_s \lambda \Psi}\}_{s=1}^{t-1}$ respectively. Therefore, from $u^{t+1}$ to obtain $w^{t+1}$, we should use $\mathbf{prox}_{\eta_t \lambda \Psi}$ to balance $-\eta_t g^t$.

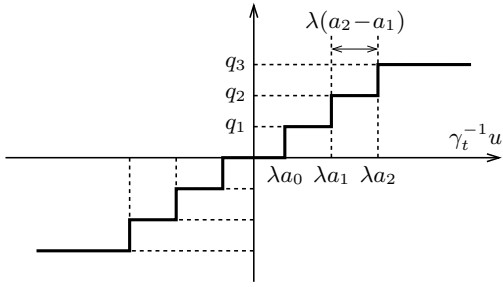

*Figure 8.* Asymptotic scaled mapping as $\gamma_t \to 0$.

**Algorithm 1** PARQ

**input:** $w^1 \in \mathbf{R}^d$, number of quantization bits $n$,
 step sizes $\{\eta_t\}_{t=1}^T$, slope schedule $\{\rho_t^{-1}\}_{t=1}^T$
**initialize:** $u^1 = w^1$
**for** $t = 1, 2, \ldots, T-1$ **do**
 $u^{t+1} = u^t - \eta_t \nabla f(w^t, z^t)$
 $\mathcal{Q}^{t+1} = \text{LSBQ}(u^{t+1}, n)$
 $w^{t+1} = \text{prox}_{\text{PARQ}}(u^{t+1}, \mathcal{Q}^{t+1}, \rho_t)$
**end for**
**output:** $w^T$

- For AProx, $u^{t+1}$ is used to accumulate $\sum_{s=1}^t \eta_s g^s$, solely contributed from $f$. Thus in computing $w^{t+1}$, we need to strike a balance with the contribution from $\lambda\Psi$ with the aggregated strength $\gamma_t = \sum_{s=1}^t \eta_s$.

While the total contributions from the forward steps $(-\eta_t g^t)$ and backward steps $(\text{prox}_{\lambda\Psi})$ are balanced in both cases, Prox-SGD spreads the backward steps on every iterate $w_t$ so the quantization effect on the last iterate eventually diminishes. In contrast, AProx always applies an aggregate proximal map to generate the last iterate, in order to balance the accumulation of pure forward steps in $u^{t+1}$.

The above interpretation highlights the importance of balance between the forward and backward steps in minimizing the sum of $f$ and $\lambda\Psi$. With the flexibility of allowing any step size rule that satisfies (10), it can be considered as a more flexible variant, or a generalization, of the regularized dual averaging (RDA) method of Xiao (2010).

Dockhorn et al. (2021) used the regularization framework and proximal maps to interpret BinaryConnect/STE and developed a generalization called ProxConnect. It is derived from the generalized conditional gradient method (Yu et al., 2017), through the machinery of Fenchel-Rockafellar duality. We derived AProx as an direct extension of RDA (Xiao, 2010), but realized that it is indeed equivalent to ProxConnect, with some minor differences in setting $\gamma_t$. Nevertheless, our construction through balancing forward and backward steps provides a more intuitive understanding of the algorithm and may shed light on further development of structure-inducing optimization algorithms.

### 3.2. Convergence Analysis

To simplify the presentation, we define

$$F_\lambda(w) := \mathbf{E}_z[f(w, z)] + \lambda\Psi(w).$$

The following theorem concerns the convergence of AProx in terms of the weighted average $\bar{w}^t = \frac{1}{\sum_{s=1}^t \eta_s} \sum_{s=1}^t \eta_s w^s$. This result appeared in Dockhorn et al. (2021, Cor. 5.2.). We include it here as a basis for proving last-iterate convergence and give its proof in Appendix A.1 for completeness.

**Theorem 3.1.** *Assume that $f(w, z)$ is convex in $w$ for any $z$, $\Psi$ is convex, and $F_\lambda$ is continuous with Lipschitz constant $G$. Also, let $\mathcal{W}^\star$ be the set of minimizers of $F_\lambda(w)$. Then,*

(a) *If the stepsize $\eta_t$ satisfies (10) and $\{w_s\}_{s=1}^t$ are generated by algorithm (11), then the weighted average $\bar{w}^t$ converges in expectation to a point in $\mathcal{W}^\star$.*

(b) *Let $w^0$ be an initial point, $R = \min_{w^\star \in \mathcal{W}^\star} \|w^0 - w^\star\|_2$ and the step size $\eta_t = \frac{R}{2G}\sqrt{\frac{1}{t}}$, then*

$$\mathbf{E}\big[F_\lambda(\bar{w}^t)\big] - F_\lambda(w^\star) \leq GR\frac{2 + 1.5\ln(t)}{\sqrt{t}},$$

*where the expectation $\mathbf{E}[\cdot]$ is taken with respect to the sequence of random variables $\{w^1, \ldots, w^t\}$.*

While convergence results on the averaged iterates $\bar{w}^t$ are conventional in the stochastic optimization literature, they are far from satisfactory for QAT. In particular, the averaged iterates $\bar{w}^t$ are most likely *not* quantized even if every iterate $w^t$ is quantized. Therefore, only the last iterate is meaningful for QAT in practice.

In general, last-iterate convergence of stochastic/online algorithms is crucial for regularized optimization problems aiming for a structured solution (such as sparsity and quantization). Here we establish last-iterate convergence of AProx.

**Theorem 3.2** (Last-iterate convergence of AProx for convex optimization)**.** *Under the same assumptions as in Theorem 3.1, the last iterate $w^t$ of AProx satisfies*

$$\mathbf{E}\big[F_\lambda(w^t)\big] - F_\lambda(w^*) \leq GR\frac{2 + 1.5\ln(t)}{\sqrt{t}}.$$

The proof of Theorem 3.2 is provided in Appendix A.2. We note that this convergence rate matches the average-iterate convergence rate established in Theorem 3.1.

We note that AProx updates the variable $u^t$ with a simple SGD step. In practice, replacing it with more sophisticated methods such as Adam (Kingma & Ba, 2014) or AdamW (Loshchilov & Hutter, 2018) gives better performance. We leave their convergence analysis for future work.

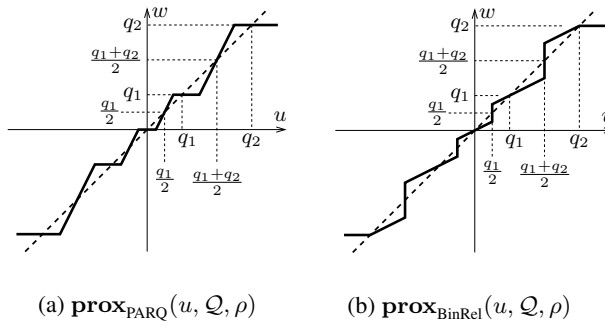

(a) $\mathbf{prox}_{\mathrm{PARQ}}(u, \mathcal{Q}, \rho)$      (b) $\mathbf{prox}_{\mathrm{BinRel}}(u, \mathcal{Q}, \rho)$

*Figure 9.* Proximal maps of PARQ and BinaryRelax.

## 4. PARQ: A Practical Implementation

A practical issue for implementing AProx with PAR is how to choose the PAR parameters $\{q_k\}_{k=1}^{m}$ and $\{a_k\}_{k=0}^{m-1}$, as well as the regularization strength $\lambda$; see their roles in the proximal map in Figure 7. In particular, $\{q_k\}$ are the target quantization values for $w^t$ and $\lambda$ and $\{a_k\}$ determine the quantization thresholds on the scaled input $\gamma_t^{-1}u^t$. In practice, it is very hard to choose these parameters a priori for different models and datasets. Therefore, we propose a heuristic approach to estimate the target values $\{q_k\}$ online and at the same time avoid setting $\lambda$ and $\{a_k\}$ explicitly.

Given a vector $u^t \in \mathbf{R}^d$, we need to quantize it (element-wise) to a vector $w^t \in \mathcal{Q}^d$ where $w_i^t \in \mathcal{Q}$ for $i = 1, \ldots, d$. We use the least-squares binary quantization (LSBQ) approach (Pouransari et al., 2020) to estimate the target quantization values in $\mathcal{Q}$. LSBQ employs a form of $n$-bit *scaled binary quantization*. Specifically, let

$$w_i = \textstyle\sum_{j=1}^{n} v_j s_j(u_i),$$

where the $v_j$'s satisfy $v_1 \geq \cdots \geq v_n \geq 0$ and each $s_j : \mathbf{R} \to \{-1, 1\}$ is a binary function. The optimal $\{v_j, s_j(\cdot)\}_{j=1}^{n}$ for approximating $u \in \mathbf{R}^d$ in the least-squares sense can be found by solving the problem:

$$\begin{aligned} \text{minimize}_{\{v_j, s_j(\cdot)\}} \quad & \textstyle\sum_{i=1}^{d}\big(u_i - \sum_{j=1}^{n} v_j s_j(u_i)\big)^2 \\ \text{subject to} \quad & v_1 \geq v_2 \geq \cdots \geq v_n \geq 0, \\ & s_j : \mathbf{R} \to \{-1, 1\}, \; j = 1, \ldots, n. \end{aligned}$$

For $n = 1$ (1-bit quantization), the solution is well-known:

$$v_1 = \|u\|_1 / d, \quad \text{and} \quad s_1(u_i) = \mathrm{sgn}(u_i);$$

see, e.g., Rastegari et al. (2016). Pouransari et al. (2020) derived the solutions for the $n = 2$ case and the ternary case ($n = 2$ with $v_1 = v_2$). For $n > 2$, there is no closed-form solution, but Pouransari et al. (2020) gives a simple greedy algorithm for *foldable* representations, which satisfy

$$s_j(u_i) = \mathrm{sgn}(u_i - \textstyle\sum_{\ell=1}^{j-1} v_\ell s_\ell(u_i)), \quad j = 1, \ldots, n.$$

This is the scheme that we adopt in PARQ.

*Table 1.* ResNet test accuracy on CIFAR-10. Full-precision (FP) accuracy is shown in parentheses under each model depth.

| Depth | # bits | STE | BinaryRelax | PARQ |
|---|---|---|---|---|
| 20
(91.82) | 1 | 89.56 ±0.18 | 89.98 ±0.13 | **90.48** ±0.26 |
| | T | 90.94 ±0.15 | 91.25 ±0.07 | **91.45** ±0.11 |
| | 2 | 91.22 ±0.15 | 91.57 ±0.06 | **91.71** ±0.03 |
| | 3 | 91.84 ±0.22 | 91.77 ±0.05 | **91.97** ±0.04 |
| | 4 | 91.93 ±0.04 | 91.92 ±0.16 | 91.93 ±0.05 |
| 56
(93.08) | 1 | 91.55 ±0.33 | 91.75 ±0.37 | 91.47 ±0.35 |
| | T | 92.42 ±0.09 | 92.34 ±0.23 | **92.97** ±0.15 |
| | 2 | 92.72 ±0.27 | 92.30 ±0.40 | 92.77 ±0.10 |
| | 3 | 92.73 ±0.44 | 92.86 ±0.40 | 92.86 ±0.25 |
| | 4 | 92.34 ±0.23 | 92.59 ±0.10 | **92.78** ±0.30 |

Once a set of (exact or approximate) solution $\{v_j\}_{j=1}^{n}$ is obtained, the resulting quantization values can be written in the form $\pm v_1 \pm \cdots \pm v_n$ by choosing either $+$ or $-$ between the adjacent operands. For example, the largest and smallest values in $\mathcal{Q} = \{\pm q_1, \ldots, \pm q_m\}$ are $q_m = v_1 + \cdots + v_n$ and $-q_m = -v_1 - \cdots - v_n$. Since there are $n$ binary bits, the total number of target values is $|\mathcal{Q}| = 2^n$.

The selection of $\{a_k\}$ and $\lambda$ is somewhat arbitrary and not consequential. We can choose them so that the asymptotic graph in Figure 8 matches the hard-quantization map depicted in Figure 1. That is, we can let $\lambda a_k = (q_k + q_{k+1})/2$, but never really use them once $\mathcal{Q}$ is found by LSBQ.

While in theory we require $\gamma_t = \sum_{s=1}^{t} \eta_s \to +\infty$, in practice $\gamma_t$ does not become very large due to the finite number of iterations we run with diminishing step sizes. Therefore, its effect on scaling the horizontal axis in Figures 7 and 8 is limited and can be absorbed by tuning the step size. On the other hand, we would like the proximal map to be able to converge to hard-quantization by the end of training (so we have fully quantized solutions). For this purpose, we use an independent schedule for growing the slope of the slanted segments. Specifically, we emulate the proximal map in Figure 7 with the one in Figure 9(a), where $\mathcal{Q}$ is calculated from LSBQ, and $\rho$ is the slope of the slanted segments. For convenience, we specify a schedule for the *inverse slope* $\rho_t^{-1}$ to vary monotonically from 1 to 0 during $T$ steps of training (so the slope $\rho_t$ go to infinity). For example,

$$\rho_t^{-1} = \frac{1}{1 + \exp\left(s(t - t_1)\right)}, \tag{13}$$

where $s > 0$ is the steepness parameter, and $t_1$ is the transition center (usually $t_1 = T/2$). This schedule changes $\rho_t^{-1}$ roughly from 1 to 0, taking value 0.5 at the transition center $t_1$. The steepness parameter $s$ controls how fast the transition from 1 to 0 happens, with larger $s$ corresponding to steeper transitions.

Putting everything together, we have PARQ in Algorithm 1.

*Table 2.* Quantized ResNet-50 test accuracy on ImageNet.

| Depth | # bits | STE | BinaryRelax | PARQ |
|---|---|---|---|---|
| 50 (75.60) | 1 | 66.17 ±0.04 | 66.14 ±0.28 | **66.71** ±0.13 |
| | T | 70.94 ±0.19 | 71.59 ±0.11 | 71.45 ±0.11 |
| | 2 | 72.38 ±0.10 | 72.64 ±0.17 | 72.71 ±0.19 |
| | 3 | 73.58 ±0.09 | 74.02 ±0.09 | 73.94 ±0.10 |
| | 4 | 74.52 ±0.04 | 74.58 ±0.04 | **74.83** ±0.19 |

*Table 3.* Quantized DeiT test accuracy on ImageNet.

| Size | # bits | STE | BinaryRelax | PARQ |
|---|---|---|---|---|
| Ti (71.91) | 1 | 51.62 ±0.18 | 52.62 ±0.03 | **55.43** ±0.23 |
| | T | 61.43 ±0.08 | 62.18 ±0.11 | 62.32 ±0.28 |
| | 2 | 64.81 ±0.15 | 65.20 ±0.04 | **66.60** ±0.18 |
| | 3 | 69.02 ±0.11 | 69.26 ±0.03 | **69.60** ±0.22 |
| | 4 | 70.95 ±0.11 | 71.06 ±0.09 | 71.21 ±0.11 |
| S (79.80) | 1 | 70.07 ±0.03 | 70.69 ±0.07 | **73.40** ±0.19 |
| | T | 75.83 ±0.06 | 76.02 ±0.03 | **76.74** ±0.06 |
| | 2 | 77.40 ±0.01 | 77.43 ±0.04 | **77.94** ±0.04 |
| | 3 | 79.02 ±0.14 | 79.11 ±0.07 | 79.04 ±0.04 |
| | 4 | 79.57 ±0.04 | 79.55 ±0.12 | 79.61 ±0.04 |
| B (81.73) | 1 | 78.79 ±0.03 | 79.02 ±0.03 | **79.35** ±0.04 |
| | T | 80.50 ±0.01 | 80.61 ±0.08 | 80.62 ±0.01 |
| | 2 | 80.73 ±0.17 | 80.81 ±0.14 | 80.97 ±0.20 |
| | 3 | 80.54 ±0.20 | 80.94 ±0.05 | **81.49** ±0.13 |
| | 4 | 80.45 ±0.10 | 80.76 ±0.12 | **81.60** ±0.12 |

## 5. Experiments

We train quantized convolutional and vision-transformer models using QAT on image classification tasks across five bit-widths: ternary (T) and 1–4 bits. For each model and bit-width pair, we compare PARQ with two existing QAT methods: STE/BinaryConnect (Courbariaux et al., 2015) and BinaryRelax (Yin et al., 2018).

Specifically, STE/BinaryConnect uses the hard-quantization map in Figure 1, PARQ applies the proximal map in Figure 9(a) with slope annealing, and BinaryRelax effectively uses the proximal map in Figure 9(b) where the slope of slanted segments gradually decreases to 0. We note that $\text{prox}_{\text{PARQ}}$ is the proximal map of a convex PAR, but STE and $\text{prox}_{\text{BinRel}}$ do not correspond to convex regularization.

Each entry in Tables 1–3 shows the mean and standard deviation of test accuracies over three randomly seeded runs.

### 5.1. ResNet on CIFAR-10

We first evaluate quantized ResNet-20 and ResNet-56 (He et al., 2016) on CIFAR-10. All weights, including those in the final projection layer, are quantized. We train for 200 epochs using SGD with 0.9 momentum and 2e−4 weight decay. Following Zhu et al. (2022), the 0.1 learning rate decays by a factor of 10 at epochs 80, 120, and 150.

As shown in Table 1, PARQ performs competitively to STE and BinaryRelax across all bit-widths. For 1-bit ResNet-20, it outperforms STE by nearly one accuracy point. It is the only QAT method for ternary ResNet-56 reaching within ∼0.1 points of full-precision accuracy.

### 5.2. ResNet on ImageNet

For QAT of ResNet-50 (He et al., 2016) on ImageNet, we quantize all residual block weights per channel by computing $\mathcal{Q}$ row-wise over tensors. We use SGD with 0.1 learning rate, 0.9 momentum, and 1e−4 weight decay. The learning rate decays by a factor of 10 every 30 epochs.

Similar to the experiments on CIFAR-10, PARQ performs capably against STE and BinaryRelax in Table 2. It shows a slight advantage in the most restrictive 1-bit case, achieving a half-point margin over the other two methods.

### 5.3. DeiT on ImageNet

Applying QAT to a different architecture, we experiment with Data-efficient image Transformers (Touvron et al., 2021, DeiT). Our DeiT experiments include the Ti, S, and B model sizes with 5M, 22M, and 86M parameters, respectively. Attention block weights are quantized channel-wise as in Section 5.2. Embeddings, layer normalization parameters, and the final projection weights are left at full precision, following the setting of Rastegari et al. (2016).

We use AdamW (Loshchilov & Hutter, 2018) to train for 300 epochs with a 5e−4 learning rate and 0.05 weight decay. We hold the learning rate at 1e−8 for the final 20 epochs (after PARQ and BinaryRelax converge to hard-quantization); this boosts performance relative to the default 1e−5 minimum. We apply RandAugment (Cubuk et al., 2020) and all prior regularization strategies (Zhang et al., 2018; Yun et al., 2019) except repeated augmentation (Berman et al., 2019).

Table 3 reveals that PARQ's performance trends persist across model sizes. For 1-bit DeiT-Ti and DeiT-S, PARQ outperforms BinaryRelax by nearly three accuracy points. PARQ also achieves the best accuracy for ternary and and 2-bit DeiT-S, as well as 3- and 4-bit DeiT-B models.

Figure 11 shows the training loss curves of three different QAT methods along with full precision (FP) training on the DeiT-Ti model. We observe that in the initial phase, PARQ closely follows the FP curve because the slope of the slanted segments in its proximal map (Figure 9(a)) is close to 1. Then the training loss of PARQ increases due to the relatively sharp transition of the slope, and it follows the STE curve closely in the second half of the training process as its proximal map converges to hard quantization. The training curve of BinaryRelax has a more gradual transition.

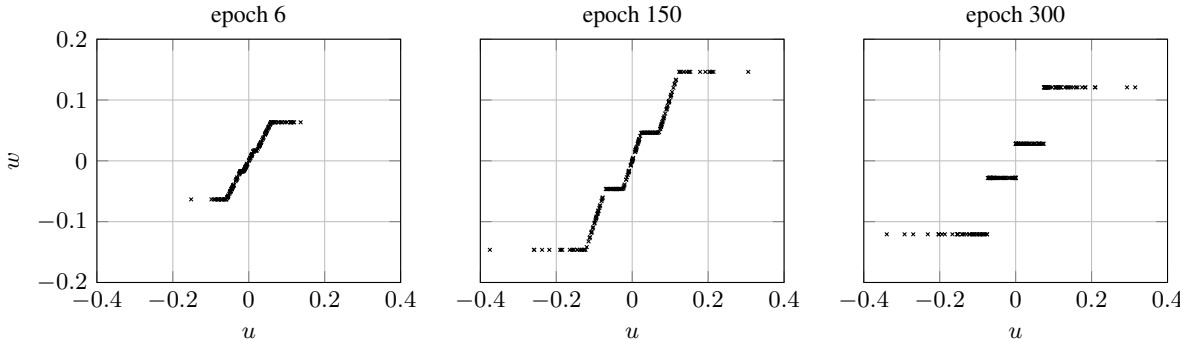

*Figure 10.* PARQ proximal maps during early, middle, and late stages of training 2-bit DeiT-Ti (value weights from an attention layer).

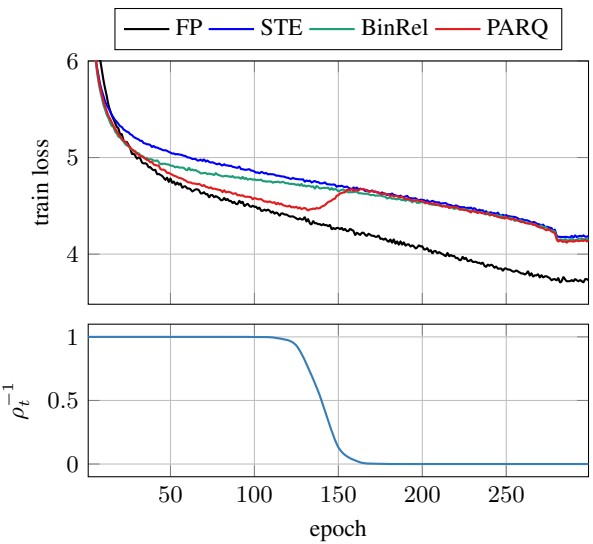

*Figure 11.* Training loss curves for 2-bit DeiT-Ti model (top) and the $\rho_t^{-1}$ schedule in (13) with $s = 50$ and $t_1 = 0.5T$.

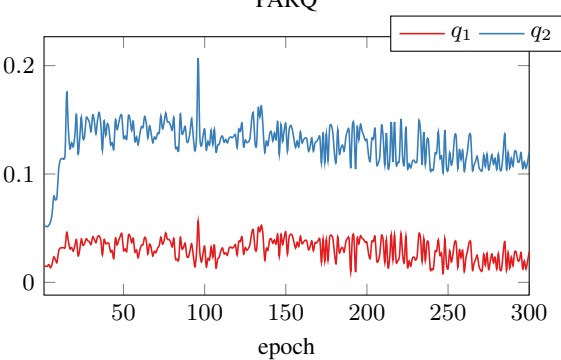

*Figure 12.* Evolution of $\{q_1, q_2\}$ estimated by LSBQ.

Figure 10 gives snapshots of how PAR gradually induces quantization in model parameters: compare the middle stage plot with Figure 9(a) and the late stage plot with Figure 1. Figure 12 shows the evolution of $\{q_1, q_2\}$ (estimated by LSBQ) during the training of a 2-bit DeiT-Ti model. They are from the same layer as the one used in Figure 10 and with the same weight initialization. It shows that both $q_1$ and $q_2$ start small from initialization, expand rapidly in the early stage of training, then slowly contract in later epochs.

Our experiments demonstrate that PARQ achieves competitive performance compared with QAT methods that correspond to using nonconvex regularization. Compared with using hard-quantization (STE) throughout the training process, the gradual evolution of PARQ from piecewise-affine soft quantization to hard quantization helps the training process to be more stable, and often converges to better local minima. This is more evident in the most demanding cases of low-bit quantization of smaller models.

## 6. Conclusion

We developed a novel optimization method for quantization-aware training (QAT) based on the framework of convex, piecewise-affine regularization (PAR). In order to avoid the diminishing regularization effect of the standard proximal SGD method, we propose an aggregate proximal (AProx) algorithm. The asymptotic form of AProx with PAR corresponds to hard quantization, thus giving a principled interpretation of the straight-through estimator (STE), which is a widely successful heuristic for QAT.

The convex regularization framework of PARQ allows the development of strong convergence guarantees. In particular, for convex loss functions, we are able to prove last-iterate convergence of the AProx method. For future work, we are interested in extending the convergence analysis for nonconvex loss functions, as well as for variants of AProx that incorporate stochastic momentum and diagonal scaling.

We have focused on PAR as an effective regularization in an optimization framework. It would also be very interesting to investigate its generalization capability in a statistical learning framework, which will help us better understand the tradeoff between model size and prediction performance.

## Impact Statement

This paper presents work whose goal is to advance the field of Machine Learning. There are many potential societal consequences of our work, none which we feel must be specifically highlighted here.

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

# A. Convergence analysis

## A.1. Proof of Theorem 3.1

We consider the framework of online convex optimization, which is more general than stochastic optimization. In particular, let $f_t = f(\cdot, z^t)$ be a function presented to us at each iteration $t = 1, 2, \ldots$, and $\Psi$ be a regularization function that we use throughout the whole process. The two-step presentation of AProx in (11) can be written in one-step as

$$w^{t+1} = \arg\min_{w \in \mathcal{W}} \left\{ \sum_{s=1}^{t} \eta_s \big( \langle g^s, w \rangle + \lambda \Psi(w) \big) + \frac{1}{2} \|w - w^0\|_2^2 \right\}, \tag{14}$$

where $w^0$ is the initial weight vector and $g^t = \nabla f_t(w^t)$. Moreover, we use a more general distance generating function $h$ to replace $(1/2) \| \cdot \|_2^2$, and define the Bregman divergence as

$$D_h(u, w) = h(u) - h(w) - \langle \nabla h(w), u - w \rangle.$$

With Bregman divergence, a more general form of AProx can be written as

$$w^{t+1} = \arg\min_{w \in \mathcal{W}} \left\{ \sum_{s=1}^{t} \big( \eta_s \langle g^s, w \rangle + \lambda \Psi(w) \big) + D_h(w, w^0) \right\}. \tag{15}$$

**Assumption A.1.** We make the following assumptions:

(a) Each loss function $f_t$ is convex and Lipschitz continuous with Lipschitz constant $G_f$.

(b) The regularizer $\Psi$ is convex and Lipschitz continuous with Lipschitz constant $G_\Psi$.

(c) The function $h$ is differentiable and strongly convex with convexity parameter $\rho$.

It follows from Assumption A.1(c) that $D_h(u, w)$ is strongly convex in $w$ with convexity parameter $\rho$.

**Theorem A.2** (Regret bound for AProx). *Under Assumption A.1, for any $w \in \mathbf{R}^d$, it holds that*

$$\sum_{s=1}^{t} \eta_s \big( f_s(w^s) + \lambda \Psi(w^s) - f_s(w) - \lambda \Psi(w) \big) \leq \frac{(G_f + \lambda G_\Psi)^2}{\rho} \sum_{s=1}^{t} 2\eta_s^2 + D_h(w, w^0). \tag{16}$$

*Proof.* We adapt the proof of Bubeck (2015, Theorem 4.3) by adding the regularizer $\Psi$ and replacing the term $h(w) - h(w^0)$ with $D_h(w, w^0)$. An advantage of this replacement is that we can use any initial point $w^0$ while the proof in (Xiao, 2010; Bubeck, 2015) requires $w^0 = \arg\min h(w)$.

Let $w^0 \in \mathbf{R}^d$ be an arbitrary initial point and define $\psi_0(w) = D_h(w, w^0)$. For $t \geq 1$, define

$$\psi_t(w) := \sum_{s=1}^{t} \eta_s \big( \langle g^s, w \rangle + \lambda \Psi(w) \big) + D_h(w, w^0).$$

The AProx algorithm (15) can be expressed as, for $t \geq 0$,

$$w^{t+1} = \arg\min_{w} \ \psi_t(w).$$

Since $D_h(w, w^0)$ is strongly convex in $w$ with convexity parameter $\rho$, the same property holds for $\psi_t$ for all $t \geq 0$. According to a basic result on minimizing strongly convex functions (e.g., Chen & Teboulle, 1993, Lemma 3.2) and the fact that $w^{t+1}$ minimizes $\psi_t$, we have

$$\psi_t(w^{t+1}) \leq \psi_t(w) - \frac{\rho}{2} \|w - w^{t+1}\|^2, \qquad t = 0, 1, 2, \ldots. \tag{17}$$

From the definition of $\psi_t$ and $\psi_{t-1}$, we have

$$\psi_t(w^t) - \psi_t(w^{t+1}) = \psi_{t-1}(w^t) - \psi_{t-1}(w^{t+1}) + \eta_t \big( \langle g^t, w^t - w^{t+1} \rangle + \lambda \Psi(w^t) - \lambda \Psi(w^{t+1}) \big). \tag{18}$$

For the left-hand side of (18), we apply (17) to obtain

$$\frac{\rho}{2}\|w^{t+1} - w^t\|^2 \le \psi_t(w^t) - \psi_t(w^{t+1}).$$

For the first term on the right-hand side of (18), we apply (17) again for $\psi_{t-1}$ to obtain

$$\psi_{t-1}(w^t) - \psi_{t-1}(w^{t+1}) \le -\frac{\rho}{2}\|w^{t+1} - w^t\|^2.$$

For the second term on the right-hand side of (18), we have

$$
\begin{aligned}
\langle g^t, w^t - w^{t+1}\rangle + \lambda\Psi(w^t) - \lambda\Psi(w^{t+1}) &\le \|g^t\|_*\|w^{t+1} - w^t\| + \lambda\Psi(w^t) - \lambda\Psi(w^{t+1}) \\
&\le G_f\|w^{t+1} - w^t\| + +\lambda G_\Psi\|w^{t+1} - w^t\| \\
&= (G_f + \lambda G_\Psi)\|w^{t+1} - w^t\|,
\end{aligned}
\tag{19}
$$

where in the first inequality we used Hölder's inequality, and in the second inequality we used Assumptions A.1(a) and A.1(b) respectively. Combining the above three inequalities with (18), we get

$$\rho\|w^{t+1} - w^t\|^2 \le \eta_t(G_f + \lambda G_\Psi)\|w^{t+1} - w^t\|,$$

which further implies

$$\|w^{t+1} - w^t\| \le \eta_t(G_f + \lambda G_\Psi)/\rho.$$

Combining this with (19) yields

$$\langle g^t, w^t - w^{t+1}\rangle + \lambda\Psi(w^t) - \lambda\Psi(w^{t+1}) \le \eta_t(G_f + \lambda G_\Psi)^2/\rho. \tag{20}$$

Next we prove that the following inequality holds for all $w \in \mathbf{R}^d$ and all $t \ge 0$:

$$\sum_{s=1}^{t}\eta_s\big(\langle g^s, w^{s+1}\rangle + \lambda\Psi(w^{s+1})\big) \le \sum_{s=1}^{t}\eta_s\big(\langle g^s, w\rangle + \lambda\Psi(w)\big) + D_h(w, w^0). \tag{21}$$

We proceed by induction. For the base case $t = 0$, the desired inequality becomes $D_h(w, w^0) \ge 0$, which is always true by the definition of $D_h$. Now we suppose (21) holds for $t - 1$ and apply it with $w = w^{t+1}$ in the first inequality below:

$$
\begin{aligned}
&\sum_{s=1}^{t}\eta_s\big(\langle g^s, w^{s+1}\rangle + \lambda\Psi(w^{s+1})\big) \\
&= \sum_{s=1}^{t-1}\eta_s\big(\langle g^s, w^{s+1}\rangle + \lambda\Psi(w^{s+1})\big) + \eta_t\big(\langle g^t, w^{t+1}\rangle + \lambda\Psi(w^{t+1})\big) \\
&\le \sum_{s=1}^{t-1}\eta_s\big(\langle g^s, w^{t+1}\rangle + \lambda\Psi(w^{t+1})\big) + D_h(w^{t+1}, w^0) + \eta_t\big(\langle g^t, w^{t+1}\rangle + \lambda\Psi(w^{t+1})\big) \\
&= \sum_{s=1}^{t}\eta_s\big(\langle g^s, w^{t+1}\rangle + \lambda\Psi(w^{t+1})\big) + D_h(w^{t+1}, w^0) \\
&\le \sum_{s=1}^{t}\eta_s\big(\langle g^s, w\rangle + \lambda\Psi(w)\big) + D_h(w, w^0), \qquad \forall\, w \in \mathcal{W}.
\end{aligned}
$$

In the last inequality above, we recognized the definition of $\psi_t$ and used the fact that $w^{t+1}$ is the minimizer of $\psi_t$. This finishes the proof of (21).

Finally we add $\sum_{s=1}^{t}\eta_s\left(\langle g^s, w^s\rangle + \Psi(w^s)\right)$ to both sides of (21) and rearrange terms to obtain

$$\sum_{s=1}^{t}\eta_s\big(\langle g^s, w^s - w\rangle + \lambda\Psi(w^s) - \lambda\Psi(w)\big) \le \sum_{s=1}^{t}\eta_s\big(\langle g^s, w^s - w^{s+1}\rangle + \lambda\Psi(w^s) - \lambda\Psi(w^{s+1})\big) + D_h(w, w^0). \tag{22}$$

For the left-hand side of (22), we use convexity of $f_s$ to obtain

$$f_s(w^s) - f_s(w) \le \langle g^s, w^s - w \rangle.$$

For the right-hand side of (22), we apply (20) to obtain

$$\sum_{s=1}^{t} \eta_s \big( \langle g^s, w^s - w^{s+1} \rangle + \lambda \Psi(w^s) - \lambda \Psi(w^{s+1}) \big) \le \frac{(G_f + \lambda G_\Psi)^2}{\rho} \sum_{s=1}^{t} \eta_s^2.$$

Combining the above three inequalities together, we have

$$\sum_{s=1}^{t} \eta_s \big( f_s(w^s) + \Psi(w^s) - f_s(w) - \lambda \Psi(w) \big) \le \frac{(G_f + \lambda G_\Psi)^2}{\rho} \sum_{s=1}^{t} \eta_s^2 + D_h(w, w^0).$$

This finishes the proof of Theorem A.2. □

Now we consider the stochastic optimization problem of minimizing $f(w) + \lambda \Psi(w)$ where the loss function $f(w) := \mathbf{E}_z[f(w, z)]$. We can regard the sequence of loss functions $f_t$ in the online optimization setting as $f(\cdot, z^t)$ and compare with $w^\star = \arg\min f(w) + \lambda \Psi(w)$. In this case, the regret bound (16) becomes

$$\sum_{s=1}^{t} \eta_s \big( f(w^s, z^s) + \lambda \Psi(w^s) - f(w^\star, z^s) - \lambda \Psi(w^\star) \big) \le \frac{(G_f + \lambda G_\Psi)^2}{\rho} \sum_{s=1}^{t} \eta_s^2 + D_h(w^\star, w^0).$$

Using a standard online-to-stochastic conversion argument (e.g., Xiao, 2010, Theorem 3), we can derive

$$\sum_{s=1}^{t} \eta_s \big( \mathbf{E}\big[ f(w^s) + \lambda \Psi(w^s) \big] - f(w^\star) - \lambda \Psi(w^\star) \big) \le \frac{(G_f + \lambda G_\Psi)^2}{\rho} \sum_{s=1}^{t} \eta_s^2 + D_h(w^\star, w^0), \tag{23}$$

where the expectation $\mathbf{E}[\cdot]$ is taken with respect to the random variables $\{w^1, \ldots, w^t\}$, which in turn depends on $\{z^1, \ldots, z^t\}$.

For the ease of presentation, we denote $R^2 = \min_{w \in \mathcal{W}} D_h(w, w^0)$. Moreover, we define a weighted average of all iterates up to iteration $t$:

$$\bar{w}^t = \frac{1}{\sum_{s=1}^{t} \eta_s} \sum_{s=1}^{t} \eta_s w^s.$$

Then by convexity of $f$ and $\Psi$, we obtain

$$\mathbf{E}\big[ f(\bar{w}^t) + \lambda \Psi(\bar{w}^t) \big] - f(w^\star) - \lambda \Psi(w^\star) \le \frac{\frac{(G_f + \lambda G_\Psi)^2}{\rho} \sum_{s=1}^{t} \eta_s^2 + R^2}{\sum_{s=1}^{t} \eta_s}. \tag{24}$$

**Constant stepsize.** If the total number of iterations $T$ is known ahead of time, then we can choose an optimal constant stepsize. Let $\eta_s = \eta$ for all $s = 1, \ldots, T$, then the bound in (24) becomes

$$\frac{\frac{(G_f + \lambda G_\Psi)^2}{\rho} T \eta^2 + R^2}{T \eta} = \frac{(G_f + \lambda G_\Psi)^2}{\rho} \eta + \frac{R^2}{T \eta}.$$

In order to minimize the above bound, we take $\eta = \frac{R}{G_f + \lambda G_\Psi} \sqrt{\frac{\rho}{T}}$ and obtain

$$\mathbf{E}\big[ f(\bar{w}^T) + \lambda \Psi(\bar{w}^T) \big] - f(w^\star) - \lambda \Psi(w^\star) \le 2(G_f + \lambda G_\Psi) R \sqrt{\frac{1}{\rho T}}.$$

**Diminishing stepsize.** The right-hand side of (24) has the same form as the convergence rate bound for the classical stochastic gradient or subgradient method (e.g., Nesterov, 2004, Section 3.2.3). A classical sufficient condition for convergence is

$$\sum_{s=1}^{\infty} \eta_s = +\infty, \qquad \sum_{s=1}^{\infty} \eta_s^2 < +\infty.$$

In particular, if we take $\eta_t = \frac{R}{2(G_f + \lambda G_\Psi)} \sqrt{\frac{\rho}{t}}$, we have

$$\mathbf{E}\big[f(\bar{w}^t) + \lambda\Psi(\bar{w}^t)\big] - f(w^\star) - \lambda\Psi(w^\star) \leq (G_f + \lambda G_\Psi)R\frac{(2 + 1.5\ln(t))}{\sqrt{\rho t}}.$$

Finally, Theorem 3.1 is obtained with some simplification. In particular, if we choose the Bregman divergence as the Euclidean distance $\frac{1}{2}\|\cdot\|_2^2$, then we have $\rho = 1$. This leads to

$$\mathbf{E}\big[f(\bar{w}^t) + \lambda\Psi(\bar{w}^t)\big] - f(w^\star) - \lambda\Psi(w^\star) \leq GR\frac{(2 + 1.5\ln(t))}{\sqrt{t}},$$

where $G := G_f + \lambda G_\Psi$. This completes the proof.

### A.2. Proof of Theorem 3.2

For simplicity, we denote $F_\lambda(w) = f(w) + \lambda\Psi(w)$ and $G = G_f + \lambda G_\Psi$ where $G_f$ and $G_\Psi$ are the Lipschitz constants of $f$ and $\Psi$, respectively.

To establish the last-iterate convergence of AProx, we first introduce the following lemma, which connects the convergence of the last iteration to the convergence of the average iteration.

**Lemma A.3** (Lemma 1 in (Orabona, 2020))**.** *Given that $\{\eta_t\}_{t=1}^T$ is a non-increasing positive sequence and $\{q_t\}_{t=1}^T$ is a nonnegative sequence, the following inequality holds*

$$\eta_T q_T \leqslant \frac{1}{T}\sum_{t=1}^T \eta_t q_t + \sum_{k=1}^{T-1}\frac{1}{k(k+1)}\sum_{t=T-k+1}^T \eta_t(q_t - q_{T-k}). \tag{25}$$

Upon setting $q_t = \mathbf{E}\left[F_\lambda(w^t)\right] - F_\lambda(w^*)$ in Lemma A.3, we derive that

$$\eta_T\left(\mathbf{E}\left[F_\lambda(w^T)\right] - F_\lambda(w^*)\right) \leq \frac{1}{T}\sum_{t=1}^T \eta_t\left(\mathbf{E}\left[F_\lambda(w^t)\right] - F_\lambda(w^*)\right)$$
$$+ \sum_{k=1}^{T-1}\frac{1}{k(k+1)}\sum_{t=T-k+1}^T \eta_t\mathbf{E}\left[F_\lambda(w^t) - F_\lambda(w^{T-k})\right]. \tag{26}$$

For the first term on the right-hand side, we apply Equation 23, which yields

$$\frac{1}{T}\sum_{t=1}^T \eta_t\left(\mathbf{E}\left[F_\lambda(w^t)\right] - F_\lambda(w^*)\right) \leq \frac{G^2}{\rho T}\sum_{t=1}^T \eta_t^2 + \frac{D_h(w^*, w^0)}{T}. \tag{27}$$

To control the second term, we note that for any $1 \leq k \leq T - 1$

$$\sum_{t=T-k+1}^T \eta_t\mathbf{E}\left[F_\lambda(w^t) - F_\lambda(w^{T-k})\right] = \sum_{t=T-k}^T \eta_t\mathbf{E}\left[F_\lambda(w^t) - F_\lambda(w^{T-k})\right] \leq \frac{G^2}{\rho}\sum_{t=T-k}^T \eta_t^2. \tag{28}$$

Here we apply Equation 23 again for the last inequality upon setting $w^\star = w^{T-k}$ and use the fact that $D_h(w, w) = 0$ for all $w \in \mathcal{W}$.

Combining the above two components together, we have

$$\mathbf{E}\left[F_\lambda(w^T)\right] - F_\lambda(w^*) \leq \frac{G^2}{\eta_T\rho}\left(\frac{1}{T}\sum_{t=1}^T \eta_t^2 + \sum_{k=1}^{T-1}\frac{1}{k(k+1)}\sum_{t=T-k}^T \eta_t^2\right) + \frac{D_h(w^*, w^0)}{\eta_T T}. \tag{29}$$

**Constant stepsize.** If the total number of iterations $T$ is known ahead of time, then we can choose an optimal constant stepsize. Let $\eta_t = \eta$ for all $s = 1, \ldots, T$, then the bound in (29) becomes

$$\mathbf{E}\left[F_\lambda(w^T)\right] - F_\lambda(w^*) \leq \frac{G^2}{\rho}\left(1 + \sum_{k=1}^{T-1}\frac{1}{k}\right)\eta + \frac{D_h(w^*, w^0)}{\eta T} \leq \frac{G^2}{\rho}\left(2 + \ln(T)\right)\eta + \frac{D_h(w^*, w^0)}{\eta T}. \tag{30}$$

Here we use the fact that $\sum_{k=1}^n \frac{1}{k} \leq 1 + \ln(n)$ for all $n \geq 1$. In order to minimize the above bound, we take $\eta = \frac{1}{G}\sqrt{\frac{D_h(w^*, w^0)\rho}{(2+\ln(T))T}}$ and obtain

$$\mathbf{E}\left[F_\lambda(w^T)\right] - F_\lambda(w^*) \leq 2G\sqrt{\frac{D_h(w^*, w^0)(2 + \ln(T))}{\rho T}}. \tag{31}$$

**Diminishing stepsize.** Suppose we set the stepsize $\eta_t = \frac{\eta}{\sqrt{t}}$. Then, Equation 29 reduces to

$$\begin{aligned}
\mathbf{E}\left[F_\lambda(w^T)\right] - F_\lambda(w^*) &\leq \frac{\eta\sqrt{T}G^2}{\rho}\left(\frac{1}{T}\sum_{t=1}^T\frac{1}{t} + \sum_{k=1}^{T-1}\frac{1}{k(k+1)}\sum_{t=T-k}^T\frac{1}{t}\right) + \frac{D_h(w^*, w^0)}{\eta\sqrt{T}} \\
&\leq \frac{\eta\sqrt{T}G^2}{\rho}\left(\frac{1+\ln(T)}{T} + \sum_{k=1}^{T-1}\frac{1}{k(k+1)}\sum_{t=T-k}^T\frac{1}{t}\right) + \frac{D_h(w^*, w^0)}{\eta\sqrt{T}}.
\end{aligned} \tag{32}$$

To proceed, note that

$$\sum_{t=T-k+1}^T\frac{1}{t} \leq \int_{T-k}^T\frac{1}{t}dt = \ln\left(\frac{T}{T-k}\right) = \ln\left(1 + \frac{k}{T-k}\right) \leq \frac{k}{T-k}. \tag{33}$$

Therefore, we have

$$\begin{aligned}
\sum_{k=1}^{T-1}\frac{1}{k(k+1)}\sum_{t=T-k}^T\frac{1}{t} &= \sum_{k=1}^{T-1}\frac{1}{k(k+1)}\left(\frac{1}{T-k} + \sum_{t=T-k+1}^T\frac{1}{t}\right) \\
&\leq \sum_{k=1}^{T-1}\frac{1}{k(T-k)} \\
&= \sum_{k=1}^{T-1}\frac{1}{kT} + \sum_{k=1}^{T-1}\frac{1}{T(T-k)} \\
&= 2\sum_{k=1}^{T-1}\frac{1}{kT} \\
&\leq 2\frac{1+\ln(T)}{T}.
\end{aligned} \tag{34}$$

Invoking this result into Equation 32, we further have

$$\mathbf{E}\left[F_\lambda(w^T)\right] - F_\lambda(w^*) \leq \frac{3\eta G^2(1+\ln(T))}{\rho\sqrt{T}} + \frac{D_h(w^*, w^0)}{\eta\sqrt{T}}. \tag{35}$$

Hence, upon setting $\eta = \frac{1}{G}\sqrt{\frac{D_h(w^*, w^0)\rho}{2}}$, we derive that

$$\mathbf{E}\left[F_\lambda(w^T)\right] - F_\lambda(w^*) \leq G\left(2\sqrt{2} + \frac{3}{\sqrt{2}}\ln(T)\right)\sqrt{\frac{D_h(w^*, w^0)}{\rho T}}. \tag{36}$$

Specifically, if we choose the Bregman divergence as the Euclidean distance $\frac{1}{2}\|\cdot\|_2^2$, then we have $\rho = 1$. Upon defining $R = \min_{w^\star \in \mathcal{W}^\star}\|w^0 - w^\star\|_2$, we have

$$\mathbf{E}\left[F_\lambda(w^T)\right] - F_\lambda(w^*) \leq GR\frac{(2 + \frac{3}{2}\ln(T))}{\sqrt{T}}. \tag{37}$$

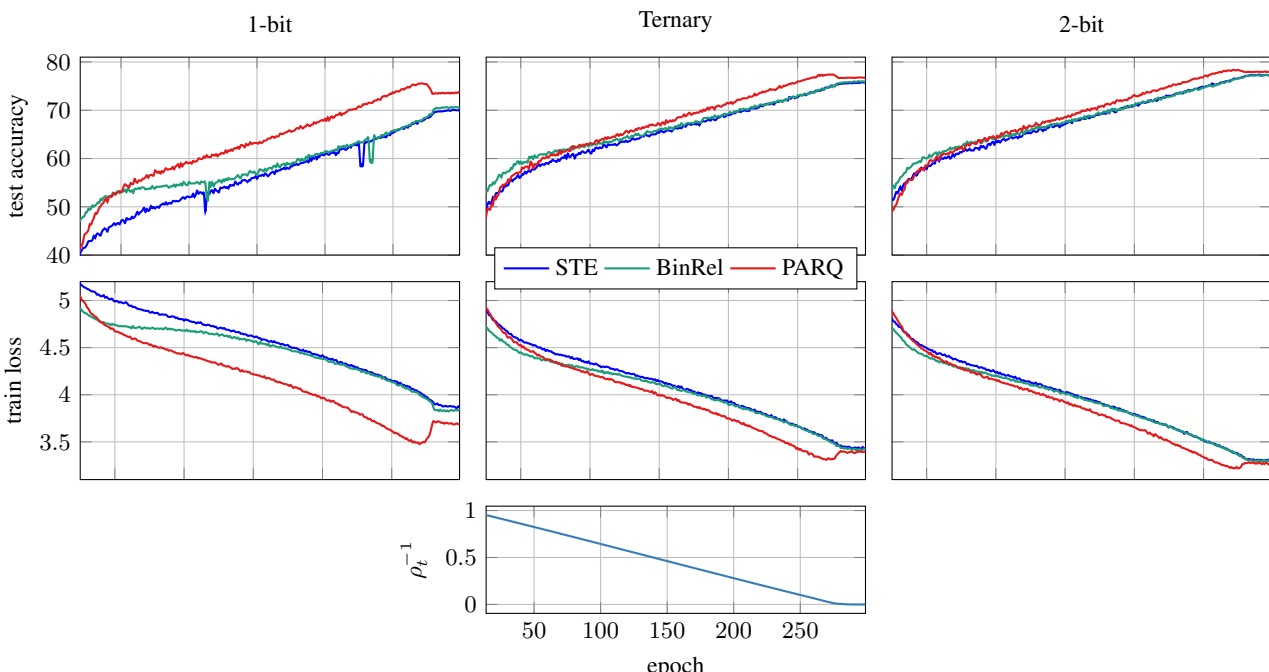

*Figure 13.* DeiT-S test accuracy (top row) and train loss (middle row) across several bit-widths (columns). All PARQ curves use a $\rho^{-1}$ schedule with $s = 1$ and $t_1 = 0.5T$ (bottom row).

## B. Additional experiment results

Figures 13–14 present accuracy and training loss curves for QAT of DeiT-S. In particular, Figure 13 uses $s = 1$, which is essentially linear during the annealing period. The 1-bit accuracy plots reveal that PARQ trains more stably than STE and BinaryRelax; it does not exhibit any sudden drops in accuracy. It performs the most consistently on DeiT-S, suggesting the relative performance of QAT methods may vary by model size.

**Ablation study on $\rho_t^{-1}$.** Table 4 shows results of 2-bit DeiT-Ti on ImageNet, trained using different $s$ (rows) and $t_1$ (columns) values in Equation (13). This sweep reveals that a shallow $s = 1$ performs best for the model and dataset setup. A later transition center of $t_1 = 0.75T$ performs noticeably better for steepness values $s \in \{10, 20\}$.

*Table 4.* Ablation of parameters in (13) on 2-bit DeiT-Ti test accuracy. The only option for $t_1$ is $0.5T$ for $s = 1$ since $\rho_t^{-1}$ decays linearly.

|       |       | $t_1$ |       |
|-------|-------|-------|-------|
|       | 0.25T | 0.5T  | 0.75T |
| 1     |       | 66.60 |       |
| 10    | 64.11 | 64.62 | 66.02 |
| 20    | 63.74 | 63.88 | 66.17 |
| 40    | 64.05 | 63.89 | 63.89 |
| 80    | 64.06 | 64.28 | 63.73 |

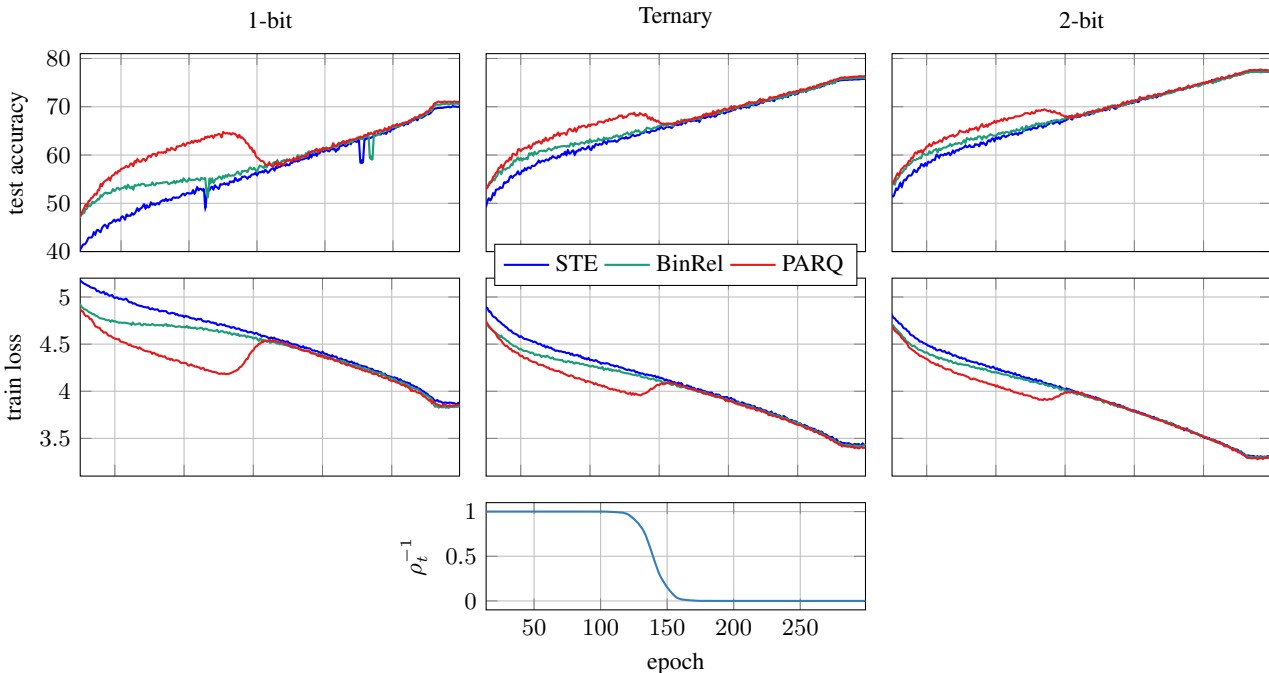

*Figure 14.* DeiT-S test accuracy (top row) and train loss (middle row) across several bit-widths (columns). All PARQ curves use a $\rho_t^{-1}$ schedule with $s = 50$ and $t_1 = 0.5T$ (bottom row).

