# OpenReview forum: "PARQ: Piecewise-Affine Regularized Quantization"
_ICML.cc/2025/Conference — ICML 2025 poster_

### Official Review · Reviewer_zbZY · 2025-03-07

**Overall Recommendation:** 3

**Summary:**

The authors propose a convex piecewise regularizer for quantization aware training. They utilize an aggregate proximal stochastic gradient method and prove that it has last-iterate convergence. They denote that their method is equivalent to a previously proposed ProxConnect method, however they derive their method in a different way.

## update after rebuttal
I would like to keep my score after the rebuttal period.

**Claims And Evidence:**

Yes.

**Essential References Not Discussed:**

N/A.

**Experimental Designs Or Analyses:**

Yes, the experimental design is sound.

**Methods And Evaluation Criteria:**

It makes sense. Having more benchmark datasets/tasks would have made the evaluation stronger.

**Other Comments Or Suggestions:**

Ablation studies on the choice of the $\rho_t^{-1}$ would be interesting to see.

**Other Strengths And Weaknesses:**

Strengths:
* The methodology is sound.
* The paper is well written.

Weaknesses:
* If the authors trained large language models with the proposed method, its impact would be much greater.
* Additional datasets/models/tasks would have showed wider applicability of the method.

**Questions For Authors:**

* What is behind the choice of the curve in Figure 10 for the $\rho_t^{-1}$ schedule? For example, how does an exponentially decaying schedule compare against the schedule used in the experiments?

**Relation To Broader Scientific Literature:**

Yes. Quantization aware training is a hot topic since the large language models have become expensive to serve.

**Theoretical Claims:**

Didn't check.

---

> ### Author Rebuttal · Authors · 2025-04-01
>
> We thank the reviewer for recognizing the strength of our paper (sound methodology and being well written) and giving us constructive suggestions on having more benchmark evaluations. We agree that additional empirical evaluations, especially on modern language models, will make the paper stronger. Although we do not have time to finish such experiments during the rebuttal period, we will try to include experiments on some basic language models in the final version to demonstrate its applicability.
>
> We agree that ablation study on the choice of $\rho_t^{-1}$ is very useful for better understanding the behavior of the algorithm. The particular curve in Figure 10 is of the sigmoid family. Specifically,
>
> $\rho_t^{-1} = \frac{1}{1 + \exp(s(t-t_1))}$
>
> where $t_1$ is the transition center and $s$ is the transition steepness. This schedule changes $\rho_t^{-1}$ roughly from 1 to 0 (taking value $1/2$ at the transition center $t_1$). Essentially this changes the slope $\rho_t$ from 1 to $+\infty$. Here $s>0$ is the steepness parameter: the larger $s$ is, the steeper transition it has. For example, $s=0.1$ makes the transition almost linear, and the one shown in Figure 10 is with $s=10$.
>
> Notice that the above sigmoid curve for $\rho_t^{-1}$ is essentially exponentially decaying as the reviewer suggested. Equivalently, we have the slope itself, $\rho_t$, increases exponentially.
>
> In our ablation study, we train a 2-bit DeiT-T model with PARQ. We study the choice of transition steepness $s \in \\{0.1, 1, 10, 20, 40, 80\\}$ and transition center $t_1 \in \\{0.25, 0.5, 0.75\\}$, the fraction of training progress at which transition occurs. This sweep reveals that a shallow $s = 1$ performs best for the model configuration. A later transition of $t = 0.75$ improves QAT performance for shallower steepness values $s \in \\{10, 20\\}$, while higher $s \in \\{40, 80\\}$ are relatively unaffected. We will add the above description and details of the ablation study to the appendix in the final version of the paper.
>
> | $\boldsymbol{s}$ | $\boldsymbol{t_1}$ | prec1 |
> |:---:|:---:|:---:|
> 0.1 | 0.5 | 66.48 |
> 1 | 0.5 | 66.6 |
> 10 | 0.25 | 64.11 |
> 10 | 0.5 | 64.62 |
> 10 | 0.75 | 66.02 |
> 20 | 0.25 | 63.74|
> 20 | 0.5 | 63.88 |
> 20 | 0.75 | 66.17 |
> 40 | 0.25 | 64.05 |
> 40 | 0.5 | 63.89 |
> 40 | 0.75 | 63.89 |
> 80 | 0.25 | 64.06 |
> 80 | 0.5 | 64.28 |
> 80 | 0.75 | 63.73 |

---

### Official Review · Reviewer_kqiu · 2025-03-08

**Overall Recommendation:** 4

**Summary:**

This paper proposed PARQ, a convex, piecewise-affine regularizer (PAR) for training the weights to cluster to a set of quantization points. Also, a practical implementation called PARQ is introduced. Overall, this paper has sufficient motivation, clear writing, grounded citations, and experiment enough to demonstrate their method's effectiveness.

## update after rebuttal

I don't have any further opinions.

**Claims And Evidence:**

I believe this paper's claims is supported by sufficient evidence.

**Essential References Not Discussed:**

I would like to see some comparison/discussion with [1], which also applies regularizer to circumvent STE.


[1] Towards Accurate Network Quantization with Equivalent Smooth Regularizer, ECCV2022

**Experimental Designs Or Analyses:**

Yes. I believe their experiments are reasonable.

**Methods And Evaluation Criteria:**

Yes.

**Other Comments Or Suggestions:**

Here, I would like to justify my rating from the following parts and sincerely hope the author could modify their paper accordingly.

Motivation and contribution to the field: STE is a widely used method to avoid the non-differentiable rounding operation in quantization. Also, it does incur some issues such as weak convergence as indicated in this paper. Thus, developing some regularization and proximal gradient methods to avoid the use of STE has been a research point here. This paper introduced a PARQ (Piecewise-Affine Regularized Quantization), which is able to serve as a regularization term to make the parameters converge to the desirable quantization levels. This provides a new contribution to theory and practice.

Writing and logical flow: In section 2, this paper first introduced Piecewise affine regularization (PAR), and then demonstrated its optimality conditions, which induce the parameter to converge to quantization levels. In section 3, this paper introduced the AProx algorithm to solve the PAR to achieve the optimal values. There are also discussions about comparing AProx and other algorithms and convergence Analysis. In section 4, this paper introduced the practical implementation of their method. Overall, a new regularization term, following a new solver algorithm and practical implementation. These make their logical flow clear.

Related to [1]. [1] is also a regularization term-based method. I consider these two papers to have the same topic. Other than that, these two papers have no similarities in method and theory, according to my understanding. I hope to see some comparison between the performance and the theory behind them.

[1] Towards Accurate Network Quantization with Equivalent Smooth Regularizer

Experimental results: Their experimental results consist of ResNet and DeiT, evaluated on widely used ImageNet. I do consider these results to be sufficient.

Novelty: The idea of convex piecewise affine regularization (PAR) and corresponding AProx solver make their method unique. To the best of my knowledge, this is the first work that proposed the regularization term from not only the practical view but also the theory view.

Theory and formula parts: I think there are two main reasons why I find it difficult to understand: one reason is that I lack professional background knowledge. Thus, I recommend having this paper reviewed by experts specifically in mathematics. The other reason is that this article lacks guidance. For example, the author should explain the purpose of each section at the beginning of each section. I believe that adding guidance will help readers clearly understand the purpose of a certain section.

**Other Strengths And Weaknesses:**

This paper is beyond my expertise. Thus, my rating is only based on the motivation, writing, logical flow, and experimental results.


I would recommend having this paper reviewed by experts specifically in theory and mathematics.

**Questions For Authors:**

No.

**Relation To Broader Scientific Literature:**

The key contributions of the paper mainly focus on the quantization area. It introduced a novel regularizer to avoid the use of STE.

**Theoretical Claims:**

No. These mathematical proofs are beyond my expertise. Thus, I only give my rating based on some reason based on some reason such as the clarity of the presentation and the logic of the arguments.

---

> ### Author Rebuttal · Authors · 2025-04-01
>
> We thank the reviewer for the overall positive feedback to our paper and especially recognizing our main novelty and contributions. Here we mainly address the reviewer’s question on the regularization approach proposed in the following reference, which we call Ref [1] hereafter.
>
> [1] Towards Accurate Network Quantization with Equivalent Smooth Regularizer, ECCV2022
>
> Ref [1] proposes smoother regularizers for inducing quantization, more specifically, of sinusoidal shape. It is clearly based on the intuition that such regularizers can help trap the weights to clusters close to a set of (evenly spaced) discrete values. This is similar to using W-shaped regularizers (which are nonsmooth) but with smooth functions in order to have a unique gradient in optimization. Unfortunately, such functions violate both properties we desired for a good regularizer for quantization: nonsmoothness and convexity (see the two paragraphs starting from Line 157 in our paper). More specifically,
> * Smooth curves like sinusoid can trap weights into separate clusters, but does not induce quantization (i.e., concentrate on discrete values). This is due to the fact that locally near the local minima, the sinusoid behaves like the squared Euclidean norm, being flat thus does not induce quantization. In contrast, nonsmooth regularizers such as $L_1$ or W-shaped or PAR are locally sharp and thus can force the weights to concentrate at the local minima. As a result, Ref[1] still needs to use additional rounding or STE steps in order to obtain discrete quantized values.
> * Convexity gives better global properties that help to avoid local minima, which explains the popularity of $L_1$ regularization over nonconvex regularizers. Without convexity (such as W-shaped regularizer), it is very hard to establish any interesting convergence theory as we do in our paper.
>
> Indeed, in the intersection of nonsmoothness and convexity, piece-wise affine function looks to be the only sensible choice. Also notice that using proximal-update of nonsmooth regularizers (instead of gradient update) avoids any problem due to the non-uniqueness of their (sub-)gradients, which is the main motivation to address with smooth regularizers by Ref[1].
>
> Finally, we agree with the reviewer that adding appropriate guidance (explaining the purpose of each section at their beginnings) will make the paper easier to read. We will be able to add them in the final paper with one extra page allowed for the main text.

---

### Official Review · Reviewer_eR92 · 2025-03-10

**Overall Recommendation:** 3

**Summary:**

they contribute a new QAT quantizer, specially optimize PAR-regularized loss functions using an aggregate proximal stochastic gradient method (AProx) and prove that it enjoys last-iterate convergence.

**Claims And Evidence:**

convincing

**Essential References Not Discussed:**

Lack some QAT quantizer

**Experimental Designs Or Analyses:**

experiment soundness.

**Methods And Evaluation Criteria:**

make sense

**Other Comments Or Suggestions:**

Since most of the current methods are STE for QAT, the current comparative experiments are reasonable, but may not be sufficient.

**Other Strengths And Weaknesses:**

This article is interesting enough, but the readability needs improvement. Some formulas lack numbers and need to be supplemented with variable explanations after the formulas.

**Questions For Authors:**

1. line 160, dist(w, Q^d) need further explanation.
2. Readability needs to be enhanced and relevant fundamentals in the field of quantization need to be supplemented, including the previous work introduced piecewise affine.
3. The regularization overhead needs to be discussed.
4. Although most existing methods use STE, I hope the authors will consider improved versions such as LSQ+.

**Relation To Broader Scientific Literature:**

Lack some QAT quantizer

**Theoretical Claims:**

theoretical correctness

---

> ### Author Rebuttal · Authors · 2025-04-01
>
> We thank the reviewer for recognizing our main contribution on the PAR regularization, the AProx method and proving its last-iterate convergence. We will work on better readability in revising the paper as suggested by the reviewer. Here are answers to the reviewer’s questions.
> 1. Line 160 the definition of $\text{dist}(w, Q^d)$ is $\text{dist}(w, Q^d)=\min_{v\in Q^d}||w-v||_2^2$, which appears in line 153.
> 2. We will add relevant fundamentals on quantization, especially on previous work using piecewise affine functions.
>   - ProxQuant introduced W-shaped regularization, which is piecewise affine but nonconvex, so it is hard to establish interesting convergence properties.
>   - Several other methods (including BinaryRelax) can be reformulated with equivalent piecewise affine regularizations, but the authors did not recognize the connection. In particular, the BinaryRelax equivalently uses the proximal map in Figure 9(b), which corresponds to a nonconvex piecewise affine regularization.
>   - Dockhorn et al (2021) focused on the piecewise affine proximal maps and made the connection with piecewise affine regularizations (PARs). But they did not realize that there exists a class of convex PARs that can lead to much stronger convergence guarantees (which is one of the main contributions of our paper).
>
> 3. We appreciate the reviewer’s comment regarding the regularization overhead. In our method, the regularization overhead is negligible compared to the cost of gradient computation. Specifically, in AProx, the additional computation arises from the proximal update described in Equation (11). However, this update leverages an explicit proximal mapping, as shown in Equation (7), which can be evaluated efficiently. In the practical implementation (PARQ), we apply LSBQ to determine quantization values by solving a constrained least-squares problem. It is important to note that this step is common across many QAT algorithms and therefore does not introduce any additional overhead specific to our method.
>
> 4. We thank the reviewer for suggesting to incorporate other quantization schemes such as LSQ+. We agree that adaptive schemes such as LSQ (Learned Step Size Quantization) and the LSQ+ extension provide more flexibility in using trainable quantization scale and offset parameters. It’s possible to replace LSBQ used in PARQ with LSQ+ for potential performance improvement, which does not impact the convex PAR theory and AProx algorithms convergence properties.
>
> We will incorporate some of the above discussions in preparing for the final submission.

---

### Official Review · Reviewer_Labs · 2025-03-11

**Overall Recommendation:** 3

**Summary:**

The paper proposes PARQ, a convex piecewise-affine regularization method for quantization-aware training. It introduces the AProx algorithm that transitions from soft to hard quantization, interprets STE as its asymptotic case, and proves last-iterate convergence.

## update after rebuttal

I confirm that I have read the author response, and would like to keep my score.

**Claims And Evidence:**

Yes.

**Essential References Not Discussed:**

No essential missing citations.

**Experimental Designs Or Analyses:**

Yes. For more details, please refer to "Other Strengths And Weaknesses"

**Methods And Evaluation Criteria:**

Yes.

**Other Comments Or Suggestions:**

No.

**Other Strengths And Weaknesses:**

Strengths:

1. Principled and Theoretical Foundation: Proposes a principled QAT method with convex regularization and proves last-iterate convergence.
2. Practical and Competitive Performance: Achieves competitive results on low-bit quantization and adaptively selects quantization values.

Weaknesses:

1. The baselines in the experiment are too old. Why is PARQ not compared with newer methods such as AdaRound[1] or N2UQ[2]?
2. Can PARQ be applied to large language models? For example, the llama Family?
3. Judging from the experimental results, the advantage of PARQ is not that great. Could the author reiterate the greatest contribution of PARQ?

[1]Nonuniform-to-Uniform Quantization: Towards Accurate Quantization via  Generalized Straight-Through Estimation

[2]Up or Down? Adaptive Rounding for Post-Training Quantization

**Questions For Authors:**

See "Strengths And Weaknesses".

**Relation To Broader Scientific Literature:**

The paper's key contributions advance quantization-aware training (QAT) by proposing a principled method using convex, piecewise-affine regularization (PAR). This builds on prior work using regularization for quantization (e.g., L1 regularization for sparsity) and proximal gradient methods. It also provides a new interpretation of the widely-used straight-through estimator (STE) as an asymptotic form of their method, bridging gaps between heuristic approaches and theoretical foundations.

**Theoretical Claims:**

Yes. For more details, please refer to "Other Strengths And Weaknesses".

---

> ### Author Rebuttal · Authors · 2025-04-01
>
> We thank the reviewer for recognizing our paper’s contributions in advancing QAT by bridging gaps between heuristic approaches and theoretical foundations. We address the reviewers comments and questions as follows:
>
> 1. “The baselines in the experiment are too old. Why is PARQ not compared with newer methods such as AdaRound[1] or N2UQ[2]?”
>
> * The baselines we use are indeed some of the early works on QAT, especially BinaryConnet/STE, which is still the de facto standard in practice. Most recent works are extensions of BinaryConnect in different ways, relying on the fundamental interpretation of “Straight-Through Estimator.” As we explained in Section 1.1, STE is a misconception we try to correct through the convex PAR framework, and we provide a more principled interpretation to it.
> * As QAT attracts more attention in recent years, there are tens of new methods proposed and published including AdaRound and N2UQ. Comprehensive comparison with recent QAT methods is not our intent in this paper, as most recent methods integrate various small additional tweaks beyond the fundamental ideas in order to boost empirical performance. And it can be unfair if we do not equip every method with similar bells and whistles. For example:
>   - N2UQ (Nonuniform-to-Uniform Quantization) introduces a particular nonuniform quantizer, it is an alternative scheme for LSBQ we use to generate the quantization targets $Q= \\{ q_1,...,q_m \\}$, which is independent of the PAR and AProx algorithm. We can replace LSBQ in PARQ (Algorithm 1) with N2UQ and reapply the rest (PAR and AProx) to test the performance, but the results are not indicative of what we care most about the effectiveness of PAR or AProx.
>   - AdaRound is actually a PTQ (post-training quantization) method, in a quite different category of QAT, see our remarks in Section 1 (second paragraph starting on Lines 39).
>
> * Among the QAT methods beyond BinaryConnect/STE, we choose to compare with BinaryRelax because it follows a similar proximal gradient approach, and the proximal map is also piecewise affine but corresponding to nonconvex regularization as shown in Figure 9(b). See also the third bullet point in our response to Reviewer DkcD's comments.
>
> 2. “Can PARQ be applied to large language models? For example, the llama Family?”
>
> * Yes, PARQ is a generic method that can be applied to train any machine learning models, including LLMs. We will try to include experiments on training a (relatively small) Llama model in the final version to demonstrate its applicability.
>
> 3. “Reiterate the greatest contribution of PARQ.”
>
> * Our main contributions include:
> Introducing a class of convex, piece-affine regularizations (PAR) that can effectively induce weight quantization;
> Developed an aggregate proximal gradient (AProx) method and proved its last-iterate convergence (first of its kind);
> Provides a principled interpretation of the widely successful heuristic of STE.
> In summary, we developed a principled approach for QAT, “bridging the gaps between heuristic approaches and theoretical foundations.”
>
> Again, as we state in the paper, the main goal of our experiments is to demonstrate that PARQ has competitive (not necessarily superior) performance against the de facto standard of BinaryConnect/STE. Indeed they become essentially the same algorithm if run for a long time, thanks to our theoretical connection. A comprehensive performance benchmark against recent QAT methods is beyond the scope of this paper, which requires taking care of many additional nuances for a fair comparison.

---

### Official Review · Reviewer_DkcD · 2025-03-23

**Overall Recommendation:** 3

**Summary:**

This paper presents a principled QAT method PARQ via convex piecewise-affine regularization (PAR). The authors examine that PAR can induce network weights to approach discrete values. Then, the paper proposes an aggregate proximal stochastic gradient method (AProx) and theoretically demonstrates its last-iterate convergence. Experiments conducted on convolutional and transformer-based models show the effectiveness of the proposed method  across five bit-widths.

## update after rebuttal
I have read the authors' rebuttal and **maintain my assessment and score** for this paper. On one hand, the authors' ​​design of convex regularization is​​ relatively novel, so **I am inclined to accept it**. On the other hand, ​​their experiments suggest that convex regularization may have limited effectiveness for non-convex deep learning loss functions, and thus it still fails to address the generalization issue in high-bit settings.​​ I acknowledge that the performances of STE and PARQ can be similar, ​​as​​ the authors ​derive​​ a generalized form of the heuristic method through theoretical analysis. However, I hope the authors can include more discussion in the final version and focus on the generalization issues of regularization in their​​ research.

**Claims And Evidence:**

The paper's arguments regarding nonsmoothness and convexity are well-founded and reasonable.

**Essential References Not Discussed:**

N/A

**Experimental Designs Or Analyses:**

1. In the experimental analyses, the authors' discussion is not comprehensive, as it primarily emphasizes the performance advantages of the 1-bit case. However, in my view, the performance of PARQ tends to degrade in experiments with more than 1 bit. For instance, in the case of 4-bit ResNet-20, PARQ exhibits a noticeable decline in accuracy. Moreover, PARQ does not achieve the best performance for more complex network. Specifically, for 1-bit ResNet-56, PARQ demonstrates the lowest accuracy and the highest standard deviation.

2. The authors lack comparisons with more recent methods, as the current analysis is limited to BinaryConnect (2015) and BinaryRelax (2018). There are several other quantization methods related to the proximal gradient method, such as Proxquant[1] and BNN++[2]. While the authors emphasize the superior performance of PARQ in the 1-bit setting, it would be more clear to include comparisons with additional methods.

Reference
[1] Bai, Yu, Yu-Xiang Wang, and Edo Liberty. "Proxquant: Quantized neural networks via proximal operators." arXiv preprint arXiv:1810.00861 (2018).
[2] Lu, Yiwei, et al. "Understanding neural network binarization with forward and backward proximal quantizers." Advances in Neural Information Processing Systems 36 (2023).

**Methods And Evaluation Criteria:**

The algorithms proposed in the paper are well-justified, and the evaluation conducted across multiple datasets for two types of models (convolutional and transformer-based) effectively reflects the models' performance.

**Other Comments Or Suggestions:**

Is there a specific reason why the best results under each setting in Tables 1-3 are not all highlighted in bold?

**Other Strengths And Weaknesses:**

This paper primarily introduces a convex regularizer that ensures discrete values and proposes an aggregate proximal map to guarantee last-iterate convergence. In my opinion, the authors' motivation and analysis for these two innovative contributions are clear and well-articulated. They effectively highlight the similarities and differences with ProxConnect, offering a distinct perspective on the problem.

**Questions For Authors:**

1. Could the authors please clarify and analyze why PARQ exhibits suboptimal performance at higher bit-widths?
2. Could the authors please clarify and analyze why PARQ demonstrates suboptimal performance in more complex networks, such as ResNet-50?
3. In Section 3.1, the title references ProxQuant, but the paragraph lacks any description or comparison related to it. Could the authors revise this section to include relevant details (does ProxQuant utilize Prox-SGD) ?

**Relation To Broader Scientific Literature:**

The paper presents an extension of the proximal gradient method, which ensures discreteness and last-iterate convergence by introducing convex piecewise-affine regularization (PAR) and the aggregate proximal stochastic gradient method (AProx). The authors also emphasize that AProx is equivalent to ProxConnect, and the straight-through estimator (STE) can be regarded as the asymptotic form of PARQ.

**Theoretical Claims:**

The authors provide a comprehensive proof for Theorems 3.1 and 3.2. I believe the last-iterate convergence of AProx is well-justified.

---

> ### Author Rebuttal · Authors · 2025-03-31
>
> We thank the reviewer for recognizing our main contributions on convex regularization for inducing quantization and the AProx method with last-iterate convergence. Our response will focus on the experiment results and analysis.
>
> We agree that we can make the discussion on experiments more comprehensive, especially with the one extra page allowed for the final version. In particular, we should emphasize that the main goal of our experiments is to demonstrate that PARQ obtains competitive performance, not necessarily always better than existing approaches such as STE or BinaryRelax (for which we provide novel, principled interpretation). There are several aspects to discuss:
> * The experiment results presented are in TEST accuracy, following the convention of the ML community. We developed PARQ as a rigorous optimization/training framework for QAT, but our work does not yet address the generalization capability of the regularization, which is an interesting topic that we aim to investigate in future work.
> * The final training losses for different algorithms are also very close, see Figures 10 and 12. Even for training loss, it is hard to guarantee that PARQ is always better than others due to the nonconvex loss functions in deep learning – the results depend on the random initializations and random mini-batches in training. Our convergence guarantees are developed for convex losses, which also imply similar behaviors around a local minimum in the nonconvex case, but in general cannot guarantee a better local minimum. Many relative comparisons in the Tables are statistically insignificant, especially given the small number of runs with random seeds.
> * On the other hand, similar performances of STE and PARQ is somewhat expected, as STE can be interpreted as the asymptotic form of PARQ. We are not contrasting two drastically different approaches, rather to argue that the one with a sound principle is as good as the widely successful heuristic (but the principled approach enables guided further investigation). Similarly, BinaryRelax uses a similar proxmap as for PARQ (see Figure 9), but does not correspond to convex PAR (hence less theory support).
> * We mainly commented on the very-low-bit cases (1 bit or ternary) due to the observed, relatively large, half-point improvements. The low-bit cases have much less freedom compared with more bits so it may be relatively easier to reveal differences between different local minima. The gradual evolution of PARQ from piecewise-affine soft quantization to hard quantization may help the training process to be more stable and more likely to converge to better local minima (See our comments in Section 6). Again there is no guarantee that this happens for every model we try (especially with a small number of trials).
>
> We agree that our experiment comparisons are limited. However, it is not our intent to give a comprehensive comparison against many recent QAT methods, as many recent methods integrate with small additional tweaks beyond the fundamental ideas in order to boost empirical performance, and it can be unfair if we do not equip every method with similar bells and whistles. We limit our comparison to BinaryConnect (STE) and BinaryRelax because they have direct connections with our method as explained above. In addition, STE is still the de facto standard in practice despite many new methods and variations proposed.
>
> * On "why the best results under each setting in Tables 1-3 are not all highlighted in bold"? We only highlight the best results that look to be statistically significant, meaning the difference between the means are apparently larger than their standard deviations.
>
> Answers to Questions for Authors:
>
> * For Questions 1 and 2, please see our itemized explanations/discussions above. In addition, we conjecture that more bit-widths and more complex networks may have the advantage of being over-parametrized, leading to very small differences between the local minima found by different methods.
>
> * For Question 3, ProxQuant (Bai et al. 2018) proposed to use the W-shaped regularizer (nonconvex) and indeed use the Prox-SGD method. As we explain in Section 3, Prox-SGD will NOT produce a meaningful quantization effect as training evolves over time ($t\to\infty$). In order to fix this empirically, as described in their implementation, the regularization parameter $\lambda$ is changed to be growing linearly $\lambda_t = \lambda \cdot t$, without principle/theory justification. It turns out that it is consistent with our AProx algorithms, where an increasing regularization strength should be applied according to our theory. Then their actual implemented algorithm is similar to PARQ, except that they use a nonconvex PAR of W-shaped. So in the end, it is more close to BinaryRelax, which we included in our experiments.

---

### Decision · Program_Chairs · 2025-05-01

**Decision:**

Accept (poster)

**Comment:**

This paper develops a principled method for quantization-aware training (QAT) based on the convex, piecewise-affine regularization. It introduces an aggregate proximal stochastic gradient method (AProx) and theoretically demonstrates its last-iterate convergence. Experiments on convolutional and transformer-based models show its effectiveness.

The main strengths of this paper are:

- the novelty of the convex regularization and the last-iterate convergence
- the connection among AProx, ProxConnect, STE and PARQ
- motivation and analysis are clear and well written

The main weaknesses of this paper are:

- the needs of more applications and comparisons like LLM
- the needs of more comprehensive discussion in the experimental analyses
- some clarification

During the rebuttal phase, all reviewers discussed, acknowledged, and provided justification.  Overall, this paper received a consistent positive recommendation, 4 weak accept and 1 accept from the reviewers.  All the concerns raised during review were addressed in the later rebuttal.
After reading all the review, the discussion and the rebuttal, the AC agrees its novelty, and its contribution to theory and practice.
As the authors agree to add more experiments, discussions and clarification based on the reviews, the AC recommend an Accept. The finial version should be revised accordingly.